

# Influence of Tidal Inundation on CO₂ Exchange between Salt Marshes and the Atmosphere

Hafsah Nahrawi[1,2], Monique Y. Leclerc[1], Gengsheng Zhang[1], Roshani Pahari[1]

[1]Atmospheric Biogeosciences Group, The University of Georgia, Griffin, 30223, USA

[2]Faculty of Resource Science and Technology, Universiti Malaysia Sarawak, Kota Samarahan, 94300, Malaysia

*Correspondence to*: Monique Y. Leclerc (mleclerc@uga.edu)

**Abstract.** Salt marshes are among the most productive and dynamic ecosystems on Earth and globally sequester an average of 210 g C m$^{-2}$ yr$^{-1}$. To understand the role of this ecosystem in the carbon cycle and its changes as a result of rapid climate change and human disturbance, a baseline record particularly on carbon dioxide ($CO_2$) exchange between this ecosystem and

atmosphere needs to be established. The goal of this study is to determine the effects tide events on the exchange of $CO_2$ in a salt marsh ecosystem dominated by *Spartina alterniflora* using the eddy-covariance method near Sapelo Island, GA. Two eddy-covariance systems were set up in July 2013 to capture 10 Hz data of $CO_2$. Results show that during daytime high tide events, a reduction of $CO_2$ exchange was observed. The conditions with a high tide to vegetation ratio had smaller $CO_2$ exchange when compared to conditions with a low tide ratio. Total daytime monthly reduction of $CO_2$ exchange for August

2014 was 15%. A greater total reduction of $CO_2$ exchange of 40% was recorded for high tide events with tide ratio of 0.75-1.0. In comparison of the effect of neap tide and spring tide on $CO_2$ exchange, neap tide days showed a greater $CO_2$ exchange as compared to spring tide days for May and October 2014, respectively. The inclusion of such results has implications to quantify the carbon budget and its changes as sea level rises.

## Keywords

Salt marsh, neap tide, spring tide, $CO_2$ exchange, eddy-covariance method

## 1 Introduction

The coastal zone covers approximately 7% of the surface of the global oceans and accounts for 14 to 30% of all oceanic primary production, 80% of organic matter burial and 90% of sedimentary mineralization. Coastal ecosystems cover about 200,000 – 400,000 km2 globally with salt marshes figuring among the most productive and dynamic ecosystems on Earth.

Globally, marshes sequester an average of 210 g C m$^{-2}$ yr$^{-1}$ (Chmura et al., 2003). Due to their effectiveness in trapping





suspended matter and associated organic C during tidal inundation the carbon sequestration contributed by vegetated coastal habitats per unit area is far greater than terrestrial forests despite of their smaller areas (Mcleod et al., 2011). Salt marsh ecosystems thus play an important role in the global carbon cycle (Duarte et al., 2005).

Salt marshes are generally located in areas receiving both inundation by sea water and exposure to air (Eleuterius

and Eleuterius, 1979). In the east coast of North America, the dominant plant species in the salt marsh area is smooth cordgrass, *Spartina alterniflora*. This plant species forms extensive monoculture stands in intertidal environments. Salt marshes inter-tidal systems are very dynamic and respond to changing environmental conditions such as changes in relative sea-level rise, tidal range, coastal engineering and human disturbance (Adam, 2002; Simas et al., 2001). With rapid changes in climate as well as increase in global temperature, current rates of sea level rise could impend salt marsh ecosystem

productivity (Kathilankal et al., 2008). In southeast Georgia, the current average sea level rise is reported at 0.3 cm yr$^{-1}$. The record is based on monthly mean sea level data from 1935 to 2013 (NOAA, 2013) (Figure 1). As a result of this sea level rise, a salt marsh will eventually change into unproductive mudflats if the marsh deposit does not keep pace with changes in sea level rise.

The increase in sea level rise as well as other development activities on the coastline areas would alter carbon

sequestration in salt marsh ecosystems. We expect sea level rise to be a chronic press driver affecting *Spartina alterniflora*. Thus, the understanding of how *Spartina alterniflora* responds to variations in inundation variability is critical to predict the future state of coastal marshes in general. However, the documentation on the exchange of $CO_2$ between salt marsh ecosystem and atmosphere measured by modern eddy-covariance systems are still very limited (Kathilankal et al., 2008).

Although few studies have been done on the exchange of $CO_2$ flux between salt marsh ecosystem and the

20 atmosphere, but there is still a lack of knowledge on how $CO_2$ exchange in this ecosystem behaves especially during the monthly neap and spring tides events. Yamamoto et al. (2009) reported that the magnitude and pattern of $CO_2$ fluxes from sediments to the atmosphere on spring and neap tide days were different. The study was conducted over a brackish-water lake dominated by Phramigmites and Juncus zone. Tong et al. (2013) also came out with similar studies using an enclosed static chamber to measure $CO_2$ flux from the Shanyutan wetland in China found similar findings. Quantifying $CO_2$ fluxes on

a diurnal scale particularly during these two main tide events helps us to understand the exchange of $CO_2$ in this ecosystem to another level.

Thus, in this study, diurnal variations of $CO_2$ flux during period with the soil exposed period and the period during which the marsh is flooded in the marsh ecosystem on selected spring and neap tide days were quantified. The effect tide on $CO_2$ exchange was also studied on a monthly basis. It is believed that the $CO_2$ exchange in intertidal salt marsh ecosystem is

30 strongly influenced by the amount of vegetation exposed to the atmosphere as well as light intensity. This paper also quantifies the amount of daytime reduction of $CO_2$ exchange. The objectives of this study are; (i) to determine the effect of



tidal inundation on $CO_2$ exchange between marsh and the atmosphere; (ii) to quantify the total amount of daytime monthly reduction of $CO_2$ exchange between the salt marsh and atmosphere.

## 2 Materials and Methods

### 2.1 Site description

5   The experiment was conducted at the Georgia Coastal Ecosystem Long Term Ecological Research (GCE-LTER) site near Sapelo Island, GA (Figure 2). It is dominated by approximately 40 ha of *Spartina alterniflora* marsh platform. The vegetation varies in height and their morphological forms are depending on location within marsh. Tall, medium and short form plants are arrayed along a gradient from the channel edge to marsh interior. The climate of the study area is characterized by strong seasonal variation and abundance precipitation. The total precipitation in 2014 were recorded at 1230

10  mm. Average annual air temperature in 2014 was 19.4°C with maximum and minimum were recorded at 35.8°C and 0.1°C respectively.

The study site is exposed to a typical semi-diurnal tide with tide height up to 2 m. During high spring tide, most of the vegetation is submerged and exposed during low spring tide and neap tide period. This condition greatly affects how $CO_2$ exchange between vegetation and atmosphere in this ecosystem (Figure 3).

### 2.2 Field measurements

Two eddy-covariance systems were installed on a flux tower established at the head of a small tidal creek on the Duplin River (latitude: 31.441°, longitude: 81.284°) in July 2013 (Figure 4). The systems were mounted at 5 m above the ground facing South (180°, hereafter south system) and North (0°, hereafter north system) respectively. The eddy-covariance systems consist of 3-D sonic anemometer (CSAT3, Campbell Scientific Inc., USA), to measure the three-wind components

and virtual air temperature, a closed-path $CO_2/H_2O$ gas analyzer (Li-7200, Li-Cor Biosciences, USA) to measure the concentration of $CO_2$ in the air at 10Hz with the control of data logger (CR3000, Campbell Scientific Inc., USA). An automated weather station with 5-min average data output was installed to measure air temperature, relative humidity, solar radiation and precipitation. Diel and tidal cycles were measured with the water level gage installed at the creek.

### 2.3 Data processing and analysis

Raw 10 Hz data from infrared gas analyzer and sonic anemometer were processed for 30 minute runs using EddyPro 6.1 software (Li-Cor Inc, Lincoln, NE, USA). Prior to running EddyPro, raw data were screened and replaced by missing value

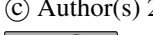


if the data has abnormal automatic gain control (AGC). The gaps in the timestamp were filled so that full timestamp for each run in EddyPro 5.2.1 software was achieved. This process was done using Matlab R2014b (The Mathworks, Inc).

Using EddyPro 6.1, The data was checked for spikes and removed according to Vickers and Mahrt (1997). A coordinate rotation using the planar fit method (Wilczak et al., 2001) was applied to the sonic anemometer data in order to remove tilt errors. Linear trend was removed. The modified Webb-Pearman-Leuning (WPL) correction for density effects due to heat and water vapor transfer (Ibrom et al., 2007) was applied to correct the calculated fluxes of $CO_2$.

Quality control was applied to each 30-min eddy-covariance data block. Data from north and south systems were combined and selected based on the climatological footprint (Kormann and Meixner, 2001; Leclerc and Foken, 2014; Leclerc and Thurtell, 1990). Only measurements that contributed to more than 70% of the $CO_2$ flux within the study area were used in this study. In 2014, a total of 37% of the observations was removed due the conditions mentioned above. Friction velocity threshold ($u^*$) for nighttime fluxes at our study site was about 0.10 ms$^{-1}$ and determined per Papale et al. (2006). Data availability and rejection due to data quality criteria is shown in Figure 5.

## 2.4 Effect of tide on $CO_2$ fluxes

To determine the effect of tide on $CO_2$ fluxes as a result of change in the amount of biomass above the water surface, the tide ratio was used. The tide ratio was calculated as:

$$Tide\ ratio = \frac{z_t}{h},$$ (1)

where $Z_t$ is the tide height and h is the mean plant height (Figure 6). When the tide ratio is equals to 1, the plants are completely submerged and the plant were completely exposed to the atmosphere when the tide ratio is equals to 0. A non-linear model using a light response curve for $CO_2$ exchange during non-flooded daytime conditions was used to estimate the reduction of $CO_2$ exchange during tide events. Thus,

$$F_{tide} = F_{mea} - F_{mod}$$ (2)

$$F_{tot} = \sum(F_{mea} - F_{mod})$$ (3)

where $F_{tide}$ is amount of $CO_2$ reduced by tide for each tide events. $F_{mea}$ is measured $CO_2$ flux. $F_{mod}$ is calculated $CO_2$ flux from a light response curve for $CO_2$ exchange model during non-flooded conditions. $F_{tot}$ is total amount of $CO_2$ reduced by tide events.

Data of August 2014 was used to study the effect of daytime tide on $CO_2$ exchange as a result of changes in biomass above the water surface as represented by the tide ratio. The effect of spring tide and neap tide events was also studied. To study the effect of tide on $CO_2$ exchange, only days with clear sky condition during spring tide and neap tide



days were used in this study. Based on the above conditions and due to limited days with clear sky particularly during spring tide days, only neap and spring tide days in May and October 2014 were selected. The selected days were about ±3 days from the exact neap and spring tide days. The field conditions during selected neap and spring tide days for May and October are shown in Figure 7. The environmental characteristics of August 2014 and days with spring tide and neap tide events in

May and October are shown in Table 1.

## 3 Results

### 3.1 Environmental characteristics of each study period

Selected environmental characteristics of each study periods are shown in Table 1. Mean air temperature for neap and spring tide days in May were $21.4 \pm 4.1°C$ and $18.7 \pm 2.3 °C$, respectively. The soil temperature for both days were $22.3 \pm 1.3 °C$

during neap tide day and $23.5 \pm 0.8 °C$ during spring tide day. Mean photosynthetic active radiation (PAR) values for neap and spring tide days were recorded at $1135 \pm 689$ µmol m$^{-2}$ s$^{-1}$ and $1127 \pm 688$ µmol m$^{-2}$ s$^{-1}$, respectively. The mean plant height in May 2014 was $0.64 \pm 0.38$ m. In Neap tide day, the highest and lowest water level were recorded at 0.79 m and -0.25 m, respectively. In Spring tide day, the highest and lowest water level measured at the creek were recorded at 1.16 m and -0.24 m, respectively. The marsh platform was not flooded during both neap and spring tide days (Figure 8).

In October, the mean air temperature for neap and spring tide days were $18.3 \pm 3.4°C$ and $17.1 \pm 3.5 °C$, respectively. The soil temperature for both days were $21.7 \pm 0.8°C$ during neap tide day and $20.3 \pm 0.6°C$ during spring tide day. Mean PAR values for neap and spring tide days were recorded at $979 \pm 557$ µmol m$^{-2}$ s$^{-1}$ and $972 \pm 488$ µmol m$^{-2}$ s$^{-1}$, respectively. In this month, the mean plant height was $0.56 \pm 0.41$ m. In neap tide day, the highest and lowest water level were recorded at 0.59 m and -0.24 m, respectively. In spring tide day, the highest and lowest water level measured at the

creek were recorded at 1.11 m and -0.23 m, respectively. No flood was observed at marsh platform during both neap and spring tide days (Figure 8).

            The tide height in August 2014 range from -0.24 to 1.48 m. The mean air and soil temperature were recorded at $27.5 \pm 2.6°C$ and $29.1 \pm 1.0°C$, respectively. Average PAR was $885 \pm 606$ µmol m$^{-2}$ s$^{-1}$ with monthly mean plant height was recorded at $0.61 \pm 0.45$ m.  The calculated tide ratios for daytime tides in August 2014 shows that, the tide ratio from 0 to

0.25 occurred most frequently with the frequency of 45% compared to other tide ratios. Meanwhile, the tide ratio between 0.75 to 1.0 occurred the least with 10% of frequency (Figure 9). The overall tide ratio for total daytime tides in August 2014 was 0.4, which indicate that 40% of the plant parts was submerged by the tides.



### 3.2 Diurnal variations of $CO_2$ flux

The carbon exchanges between salt marsh and atmosphere reached its peak from late morning to noon time which associated with high PAR level and air temperature. There is also a slight decline in $CO_2$ uptake in the afternoon from March to October. Maximum carbon fluxes were recorded in June with the maximum uptake around 10 µmol $CO_2$ m$^{-2}$ s$^{-1}$. The trends

decreased for July and August. However, the $CO_2$ exchange increased in September before it decreased again towards the end of the growing season (Figure 10). In July and August, plant suffered from high temperature during peak summer month, thus diminished the plant productivity. Respiration rates during nighttime followed the growing season pattern, increase towards the peak of the growing season and decreased afterwards.

Tidal influences on $CO_2$ exchange between the marsh and atmosphere were observed when the vegetation is

partially or submerged by the water level. It was observed that the reduction of $CO_2$ flux starts when the water level in the marsh is more than 0.2 m. The $CO_2$ exchange was reduced up to 10 times when the tidal inundation occurred compared to when there is no inundation occurred (Figure 11). This is the most ideal condition when the high tide was occurred during midday time with high PAR and temperature.

### 3.2.1 Diurnal variations of $CO_2$ flux on neap tide days

The $CO_2$ flux showed distinct diurnal variations on the neap tide days in May and October 2014 (Figure 12). In May, the maximum of $CO_2$ uptake was 8.70 µmol m$^{-2}$ s$^{-1}$ and occurred at around 1030 hours. The average of $CO_2$ flux was $1.62 \pm 0.26$ µmol m$^{-2}$ s$^{-1}$ and $-5.33 \pm 2.47$ µmol m$^{-2}$ s$^{-1}$ during nighttime and daytime, respectively. In October, the maximum of $CO_2$ uptake was observed at around noon with 6.98 µmol m$^{-2}$ s$^{-1}$. The average of $CO_2$ flux was $1.74 \pm 0.36$ µmol m$^{-2}$ s$^{-1}$ and $-3.57 \pm 1.82$ µmol m$^{-2}$ s$^{-1}$ during nighttime and daytime, respectively.

### 3.2.2 Diurnal Variations of $CO_2$ Flux on Spring Tide Days

The $CO_2$ flux also showed distinct diurnal variations on the spring tide days in May and October 2014 (Figure 13). In May, during inundation period due to spring tide, there was an obvious $CO_2$ reduction starting at 1030 to 1200 hours where it reached its peak. At night, there was also a slight reduction of $CO_2$ exchange was observed during tide event. In the daytime, during the exposed soil period between 0830 and 1630, where the uptake of $CO_2$ was the most active, the average of $CO_2$

flux was $-6.77 \pm 1.11$ µmol m$^{-2}$ s$^{-1}$. The value increased to $-4.90 \pm 1.42$ µmol m$^{-2}$ s$^{-1}$ when flooding occurred during the same period, which indicates a reduction in net uptake of $CO_2$ by the salt marsh. At night, the average of net release of $CO_2$ flux during exposed soil period was $1.42 \pm 0.02$ µmol m$^{-2}$ s$^{-1}$. During the tidal inundation period, the average of net release of $CO_2$ was $1.37 \pm 0.21$ µmol m$^{-2}$ s$^{-1}$ indicates a reduction in the average net release of $CO_2$ (Figure 13(a), Table 2).



In October, a similar pattern to May was observed. Daytime, during the exposed soil period, between 0830 to 1630, the average of $CO_2$ net uptake was $3.59 \pm 0.64$ µmol m$^{-2}$ s$^{-1}$. A reduction of the average of $CO_2$ net uptake was observed during tidal inundation period with the uptake of $2.49 \pm 0.65$ µmol m$^{-2}$ s$^{-1}$. At night, higher average of net release of $CO_2$ ($1.26 \pm 0.26$ µmol m$^{-2}$ s$^{-1}$) during the period during which the soil was exposed compared with that of the tidal inundation period ($1.11 \pm 0.63$ µmol m$^{-2}$ s$^{-1}$) (Figure 13(b), Table 2).

### 3.3 Relationship between $CO_2$ flux, mean PAR and tide ratio

The light response curve of $CO_2$ exchange describes the strong relationship between $CO_2$ flux and mean PAR for all tide ratio conditions (Figure 14). The strongest relationship was observed for non-flooded conditions ($R^2 = 0.75$), followed by the two lowest tide ratios ($R^2 = 0.7$ for each) and $R^2 = 0.6$ for the highest tide ratio ($0.75 - 1.0$). For tide ratio $0 - 0.5$, the reduction of $CO_2$ uptake started when mean PAR is approximately more than 700 µmol m$^{-2}$ s$^{-1}$. For tide ratio $0.5 - 0.75$ and 0.75 and 1.0, the reduction of $CO_2$ uptake started when mean PAR was approximately 400 µmol m$^{-2}$ s$^{-1}$.

### 3.4 Reduction of $CO_2$ exchange

As shown in Figure 10, during low tide conditions at around noon on doy 264, the maximum uptake of $CO_2$ flux was 9.39 µmol m$^{-2}$ s$^{-1}$. However, during high tide conditions at around noon on doy 255, a reduction of $CO_2$ fluxes was observed with the uptake value of 1.05 µmol m$^{-2}$ s$^{-1}$. It was observed that at water level of 0.25 m, an abrupt reduction of $CO_2$ was recorded. For monthly reduction of $CO_2$ during daytime tide events in August 2014, tide ratio of $0.75 - 1.0$ showed the highest reduction of 40% compared to the tide ratio of $0.5 - 0.75$ and $0 - 0.5$ with the reduction of 30% and 9% respectively (Table 4). The daytime total of $CO_2$ reduction in August 2014 was 15%.

## 4 Discussion

### 4.1 Influences of tide on $CO_2$ fluxes on diurnal cycles

$CO_2$ fluxes demonstrated a complex pattern with particularly when tidal inundation occurs. In May, during daytime spring tide period, the uptake of $CO_2$ was reduced by approximately 28%. In October, uptake of $CO_2$ was reduced by 31% due to spring tide events. In May, the plant height was taller than the plant height in October (Table 1). Therefore, higher percentage of plant parts were above the water surface in May compared to October. This also can be demonstrated by phenocam images taken on the site during these periods (Figure 7). In this figure, it is clearly shown that higher amount of plant parts was above the water surface during the high spring tide period in May in contrast with spring tide period in October. In a different tidal marsh ecosystem, Tong et al. (2013) reported a net release in $CO_2$ during daytime tidal





inundation period. They also reported a net uptake of $CO_2$ during nighttime tidal inundation. However, in our study, no such conditions were observed.

In May and October, the average of $CO_2$ fluxes was negative on the daily scale for both neap and spring days indicate that for both periods, the system acts as $CO_2$ sink. The difference in the average of $CO_2$ exchange in May and

5 October between neap and spring tide days was about 9% and 19%, respectively (Table 3) with higher $CO_2$ exchange was observed during neap tide days for both months. A comparison study between neap and spring tide by Tong et al. (2013) also reported similar findings. Although their findings indicate that the systems were carbon source for both spring and neap tide periods, however, neap tide day showed a greater $CO_2$ flux exchange as compared to spring tide days.

Both periods in May and October sequester $CO_2$ daily. Spring and neap tide days in October showed lower $CO_2$

uptake compared to both days in May. In May, higher photosynthesis activities during daytime corresponded with higher PAR level and plant height, as compared to October. Toward the end of the growing season, as the temperature and PAR level started to decrease the plant productivity also decreased which explain less uptake and release of $CO_2$ in October (Figure 9).

## 4.2 Influences of tide on $CO_2$ fluxes on monthly basis

Typically, tides have a strong influence on $CO_2$ exchange between marsh and atmosphere when the tide is sufficiently high to inundate the marsh. Few previous studies have shown that at a certain water table threshold, a reduction in $CO_2$ exchange between salt marsh and atmosphere was observed (Forbrich and Giblin, 2015; Kathilankal et al., 2008; Moffett et al., 2010). Site studies of these authors are dominated by marsh grass species which grow upright, either *Spartina alterniflora* (Kathilankal et al., 2008) or *Spartina foliosa* and *Distichlis spicata* (Forbrich and Giblin, 2015; Moffett et al., 2010). The

water table threshold of 0.25 m reported in this study is similar to the value reported by Kathilankal et al. (2008) and higher compared to the values of 0.17 m and 0.05 m reported by Moffett et al. (2010) and Forbrich and Giblin (2015) respectively. Both Kathilankal et al. (2008) and this study have a similar ecosystem and dominated plant species which could lead to similar pattern in the reduction of the $CO_2$ flux exchange when flooding occurs.

Our tide ratio value illustrates well the close coupling between the amount of vegetation exposed to the air when the

25 marsh is flooded with the $CO_2$ exchange between the salt marsh and atmosphere. As the tide ratio increases, the amount of vegetation exposed to the atmosphere decreases thus reduced the plant $CO_2$ net uptake from the atmosphere. Within a similar tide ratio group, the $CO_2$ net uptake increases with the increase in light intensity. This suggests that both factors (amount of biomass above the water surface and light intensity) play an important role in daytime $CO_2$ exchange in the salt marsh ecosystem. In August 2014, a reduction of the $CO_2$ flux resulting from the tide was 140.79 $\mu$mol m$^{-2}$ s$^{-1}$, corresponding to as

much as 15% of the total monthly reduction. Although this month only experienced about 10% of high tide frequency with

tide ratio 0.75 to 1.0, the reduction of approximately 40% during this period (Table 4) may slowly increase if we consider the expected of sea level rise in the coming years, thus, increase the total of monthly reduction of $CO_2$. This is because, the increase in sea level rise may lead to increase in frequency, duration of inundation and tide level, thus, lead to decrease in $CO_2$ uptake provided other factors are constant (Kathilankal et al., 2008). At the end of 21st century, the sea level rise is
expected to range between 18 and 59 cm (IPCC, 2007).

## 5 Conclusion

Salt marsh ecosystem in one of the most productive ecosystem on Earth. At Sapelo Island, GA, the plant $CO_2$ exchange was observed throughout the year with peak $CO_2$ exchange observed in June. The $CO_2$ exchange trends decrease towards the end of the year. Peak $CO_2$ exchange was observed from later in the morning to early afternoon. Similar to reduction of $CO_2$
exchange during daytime, the $CO_2$ exchange was the highest around midday correspond with high temperature and PAR level.

      A greater $CO_2$ exchange was observed during neap tide periods when compared to spring tide days for both months. Both periods in May and October sequester $CO_2$ daily. Spring and neap tide days in October showed lower $CO_2$ uptake compared to both days in May. These results suggest that the amount of biomass and the radiation load constitute the two
key factors regulating the $CO_2$ exchange in the salt marsh ecosystem daytime. For nighttime condition, only slight reduction of $CO_2$ exchange was observed during spring tide events for both days in May and October respectively.

      In August 2014, 15% of total monthly daytime reduction of $CO_2$ was observed because of tide events. Daytime high tide with tide ratio 0.75 to 1.0 contribute approximately 40% of $CO_2$ reduction, although only 10% of this event occurred during this month. With increasing of sea level rise, the frequency, duration of inundation as well as tide level may also
increase thus resulted in more $CO_2$ reduction in the coming years. Results obtained from this study would help create guidelines in sustainability practices of salt marsh ecosystems to various stakeholders such as ecosystem modelers, urban planners, realtors, homeowners as well as the tourism industry.

### Acknowledgments

The work was part of the Georgia Coastal Ecosystems LTER project and was supported by the National Science Foundation
(OCE12-37140). We also would like to thank UGA Marine Institute (contribution no. 1061) for providing us the accommodation and laboratory facilities during our field work visit at Sapelo Island, our PI, Merryl Alber and Daniela Di Iorio who supported us in many ways and provide us with great research facilities, Steve Pennings who gave a very helpful feedback on the manuscript. Finally, we would like to thank Wade Sheldon, Jacob Shalack and Timothy Montgomery and





Mohamad Widya Iskandar for helping us in various aspects of the projects particularly data management and field works. The data used in this work can be obtained from the author.

**Competing Interests**

The authors declare that they have no conflict of interest.

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





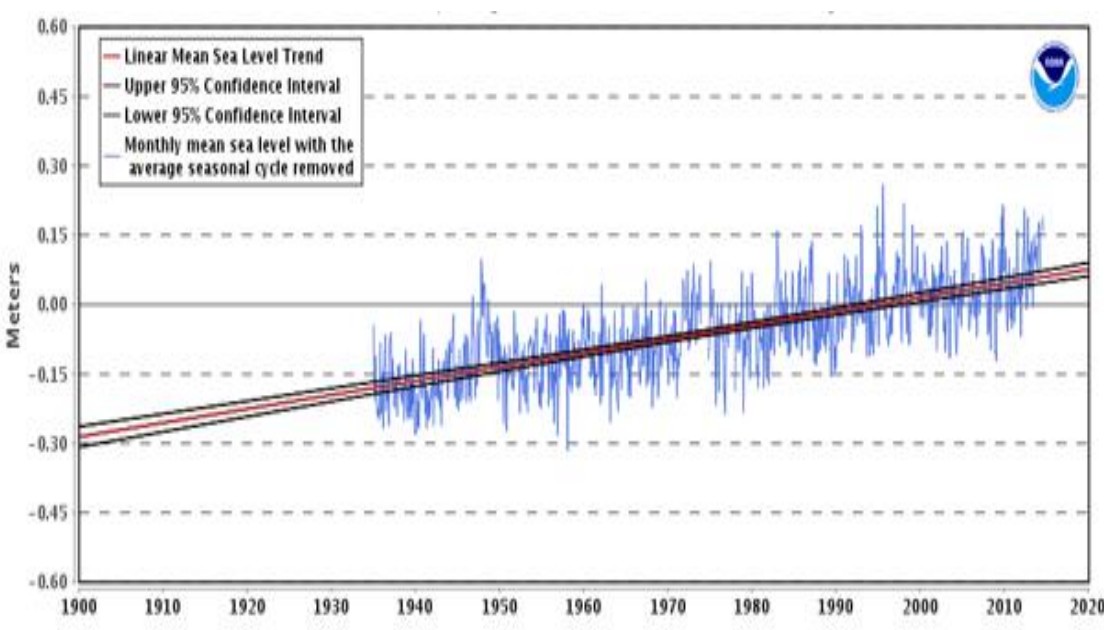

**Figure 1: Mean sea level trends, Fort Pulaski, Georgia (Source: NOAA 2013).**

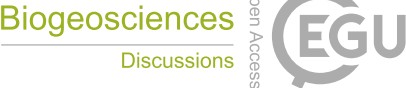

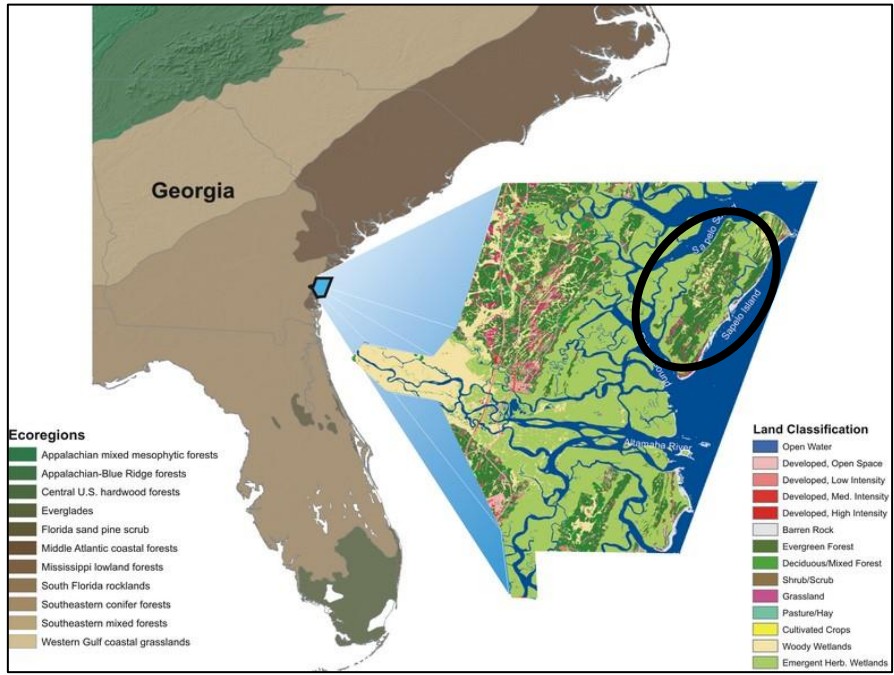

**Figure 2: Study site location (in circle) at Sapelo Island, GA (Source: gce-lter.marsci.uga.edu).**





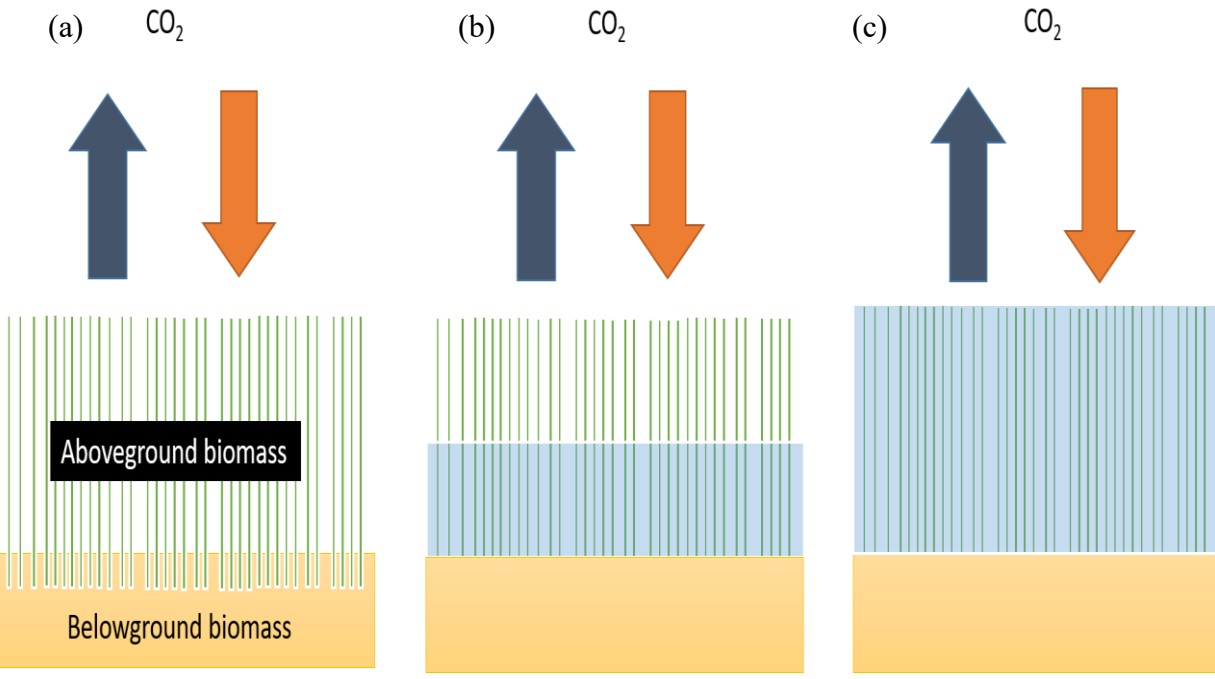

**Figure 3: Different tide level on salt marsh ecosystem affect the amount of vegetation exposed to the air; (a) Vegetation completely exposed to the air usually during low spring and neap tide (b) Vegetation is partially submerged as the water level rise (c) Vegetation completely submerged usually during high spring tide event.**



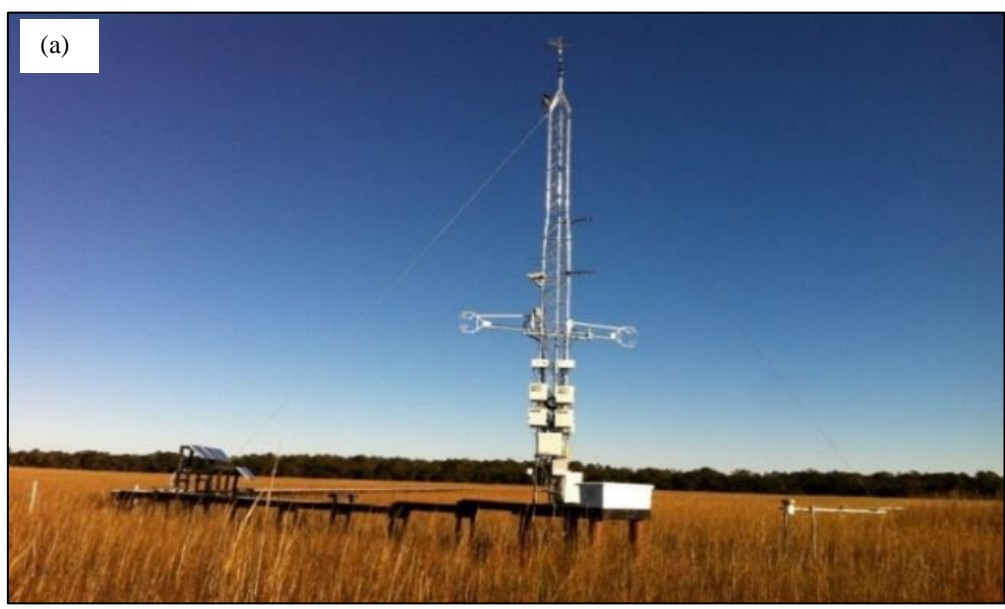

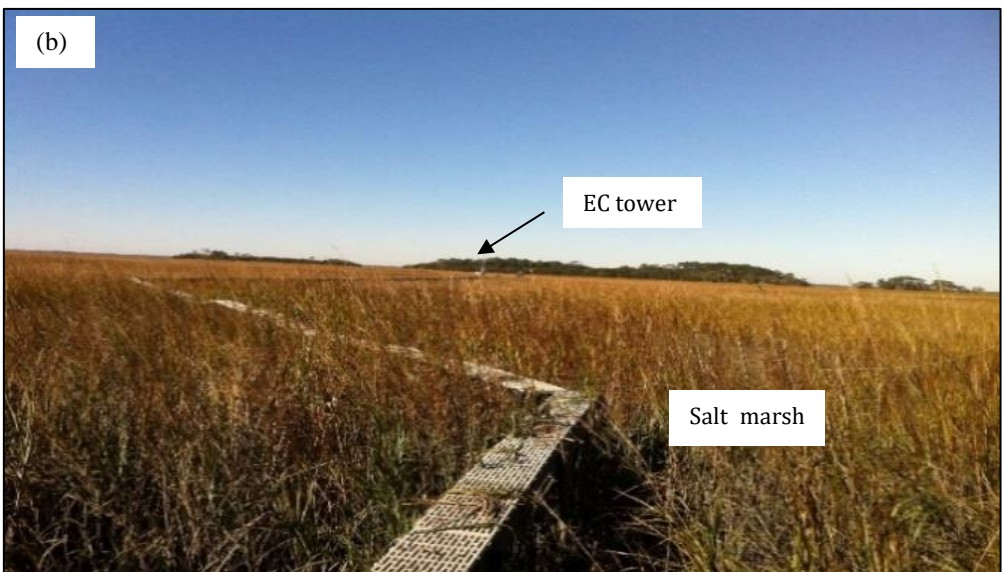

**Figure 4: (a) Eddy-covariance system installed in the marsh ecosystem; (b) View of the EC system from Duplin River.**





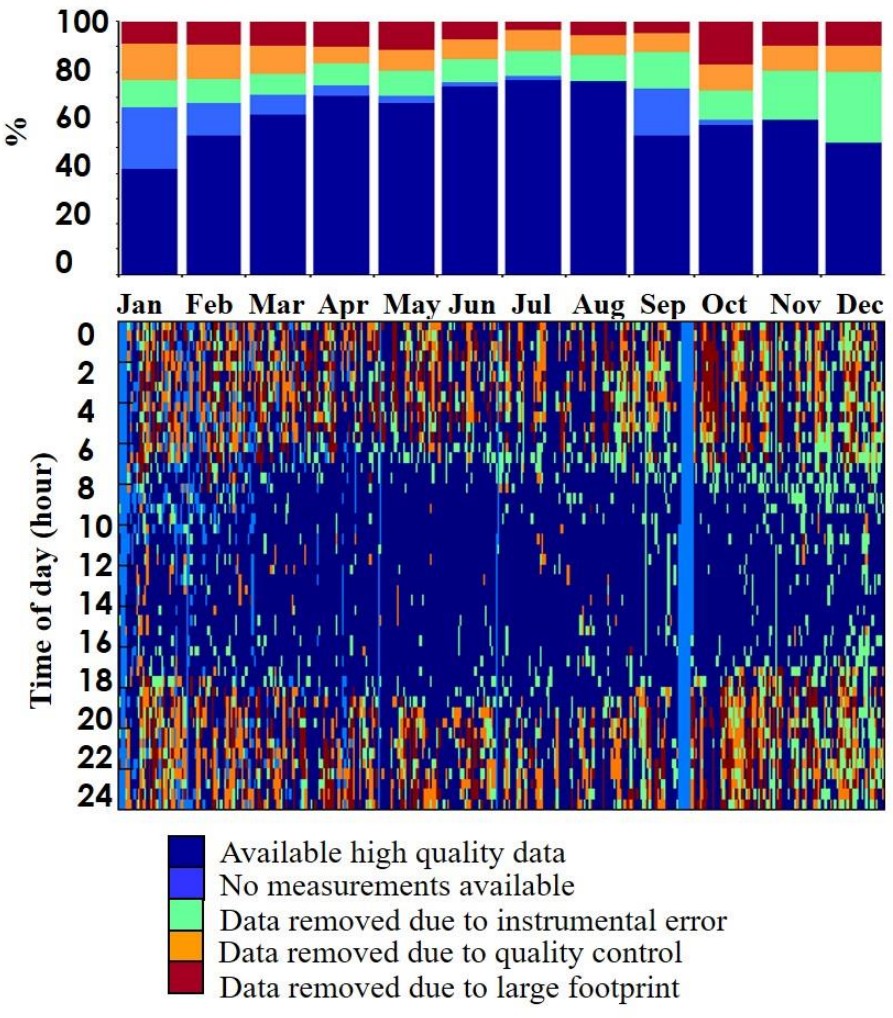

**Figure 5: The distribution of data availability and rejection for diurnal (lower panel) and monthly (upper panel) flux data in 2014.**



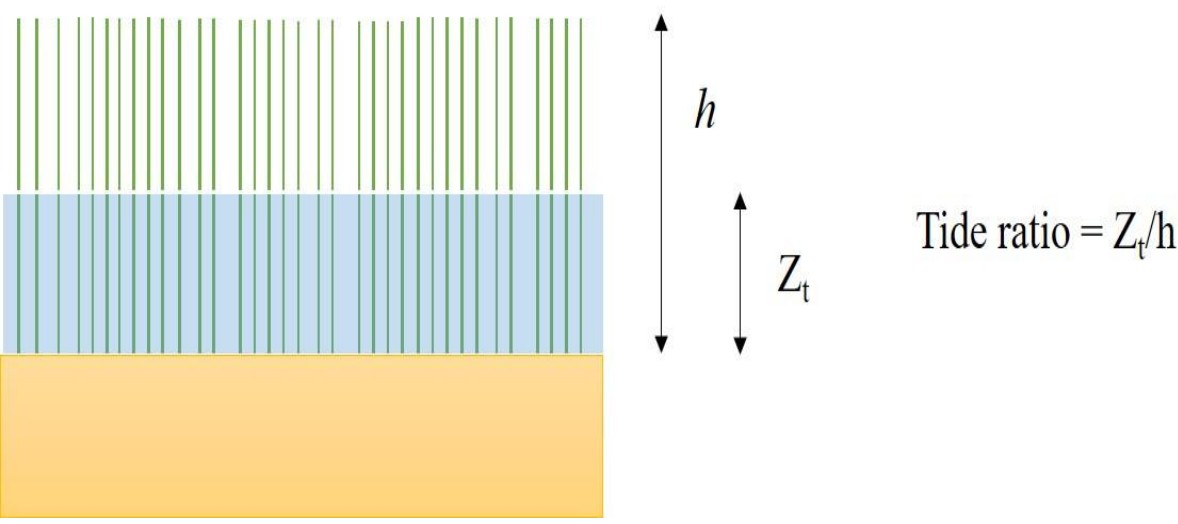

**Figure 6: Tide ratio as an indicator of amount of vegetation exposed to the atmosphere. $Z_t$ is the tide height and his monthly mean plant height.**





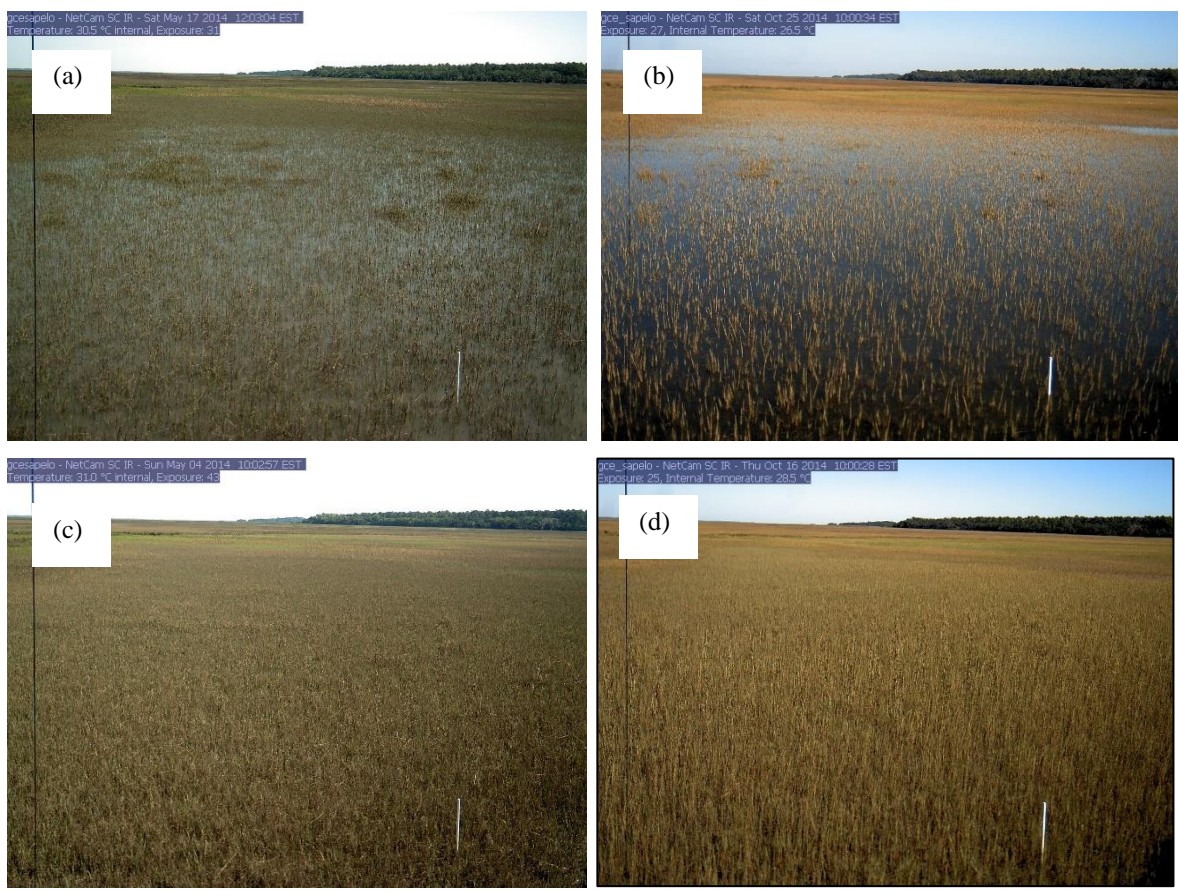

**Figure 7: Phenocam images during daytime high spring and neap tide periods. Photo taken at (a) 1200 EST on May 17, 2014 during spring tide day (b) 1000 EST on October 25, 2014 during spring tide day (c) 1000 EST on May 4, 2014 during neap tide day (d) 1000 EST on October 16, 2014 during neap tide day.**





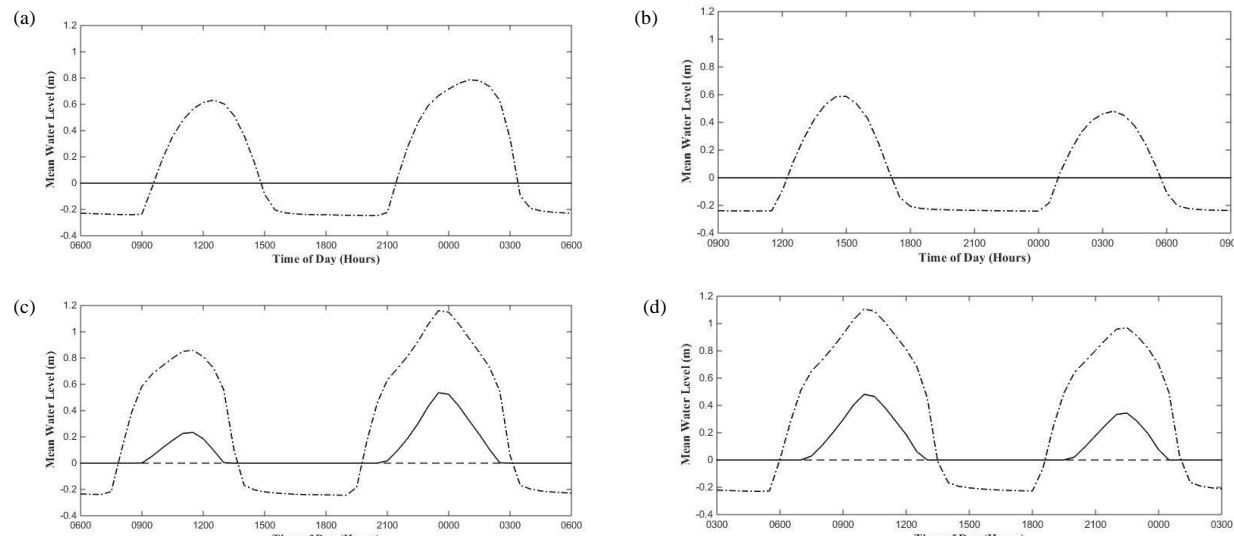

**Figure 8: Diurnal courses of tidal water height on neap tide days in (a) May 4 and (b) October 16, and spring tide days in (c) May 17 and (d) October 25. The dash-dot line represents tide height measured at the creek ((-0.23 m NAVD88) and solid line represents tie height measured at the marsh platform.**




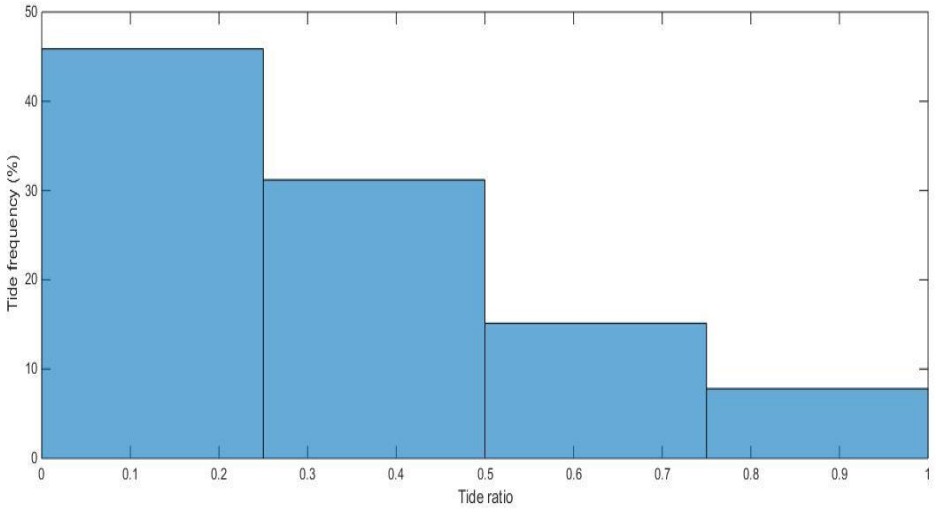

**Figure 9: Frequency of different groups of tide ratio value for August 2014.**

30

35




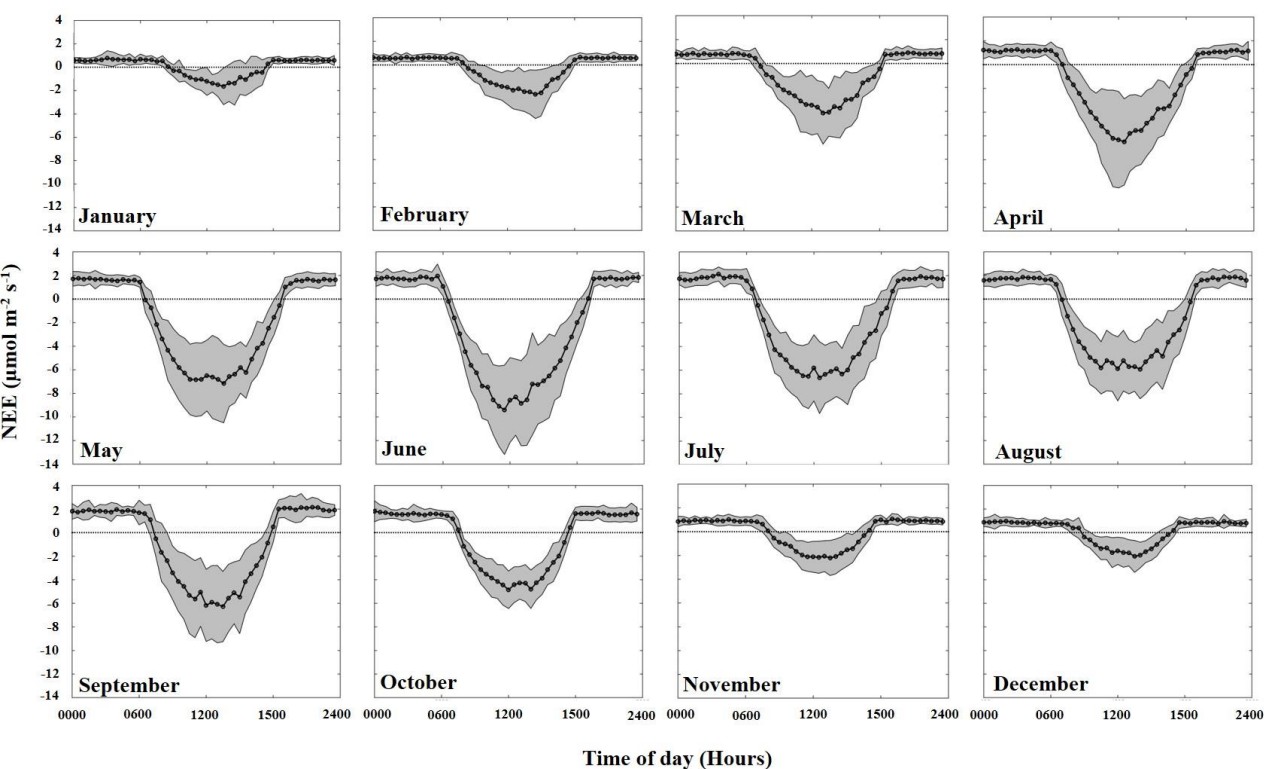

**Figure 10: Monthly average NEE for salt marsh ecosystem at Sapelo Island in 2014. The shaded areas denote the standard deviations around the mean.**




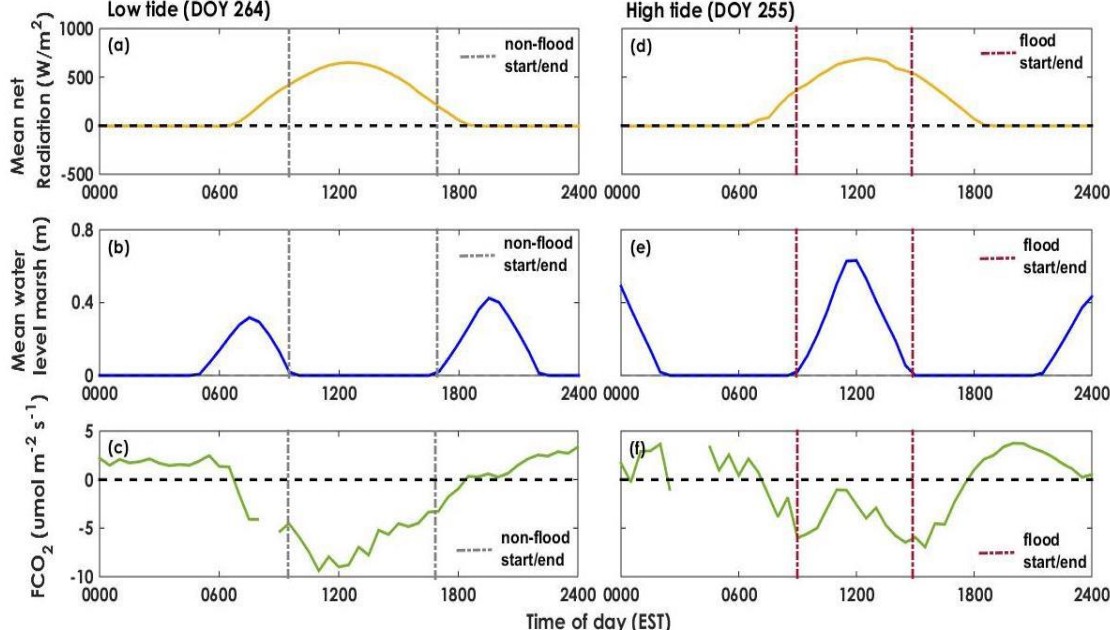

**Figure 11: CO₂ exchange when the marsh is exposed during the day (left) compared to when it is flooded (right); (top) Mean net shortwave radiation ;(middle) Mean water level on marsh; (bottom) CO₂ flux.**

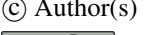



(a)
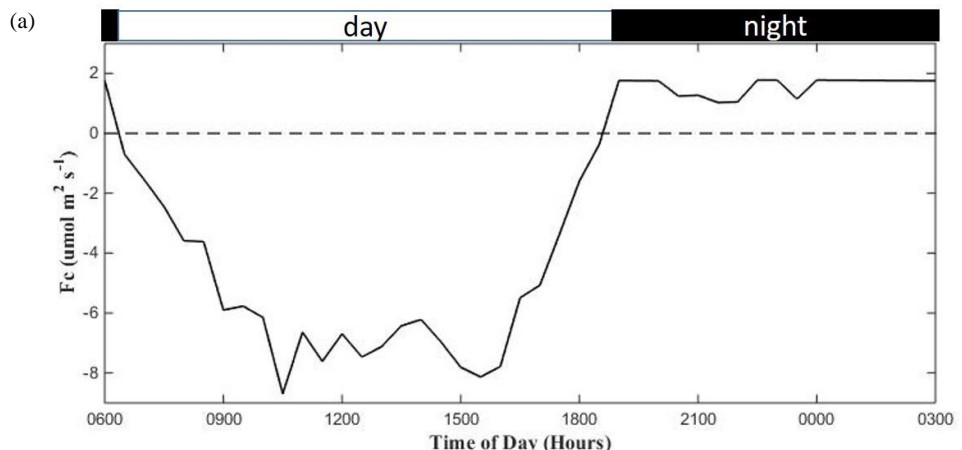

(b)
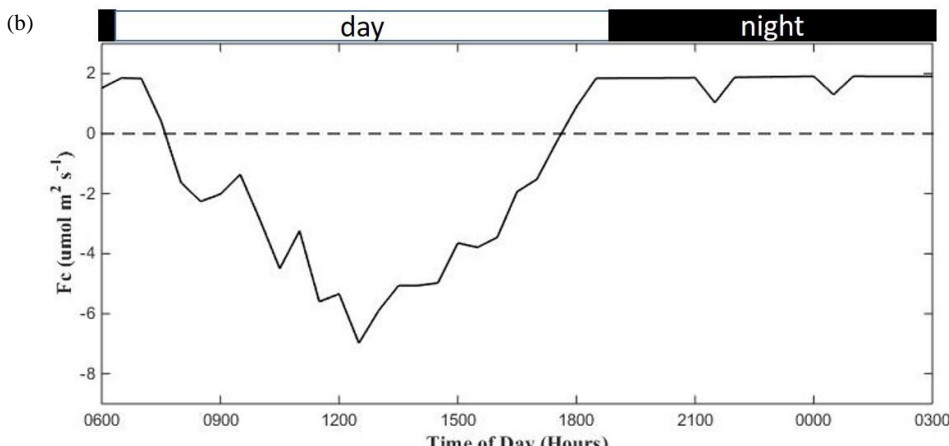

**Figure 12: Diurnal variations of CO₂ flux during neap tide days for (a) May 2014 and (b) October 2014. The negative value indicates uptake of CO₂ and positive value indicate release of CO₂.**



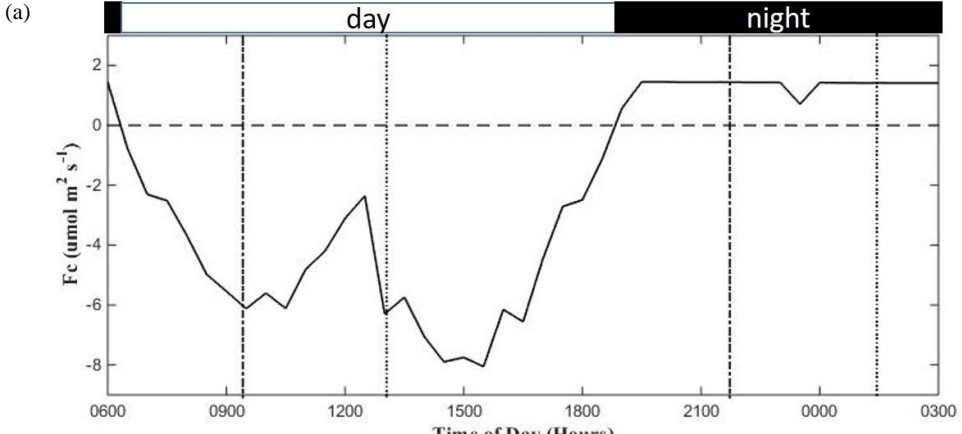

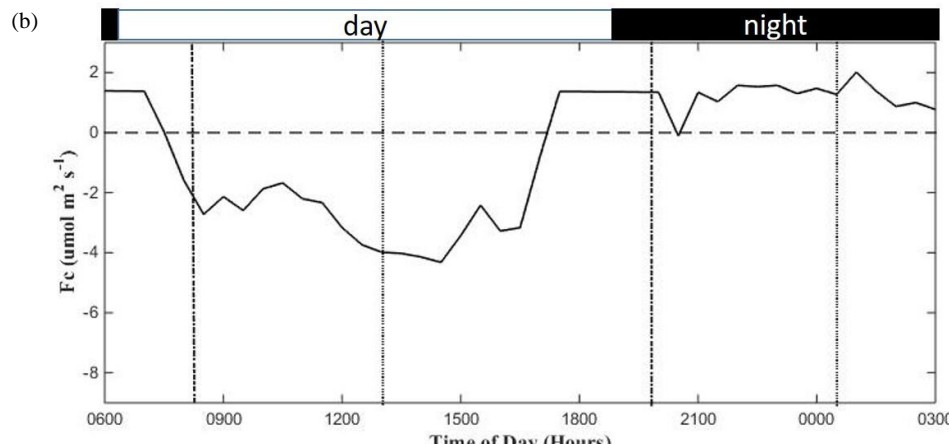

**Figure 13: Diurnal variations of CO₂ flux during spring tide days for (a) May 2014 and (b) October 2014. The negative value indicates uptake of CO₂ and positive value indicates release of CO₂. Dash-dot and dot lines represent the flood starts and ends, respectively measured at the salt marsh platform.**





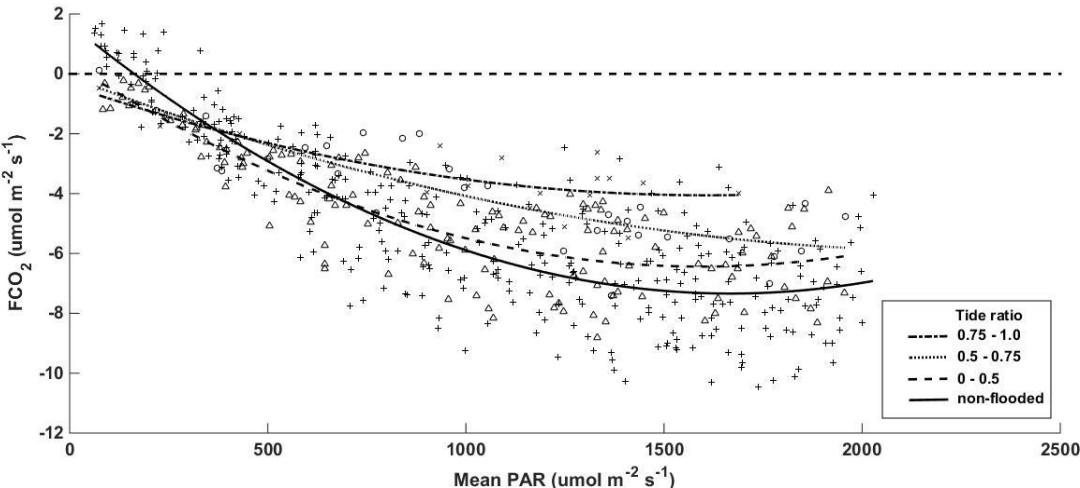

**Figure 14: Light response curve of CO₂ exchange with different levels of flooding (three tide ratios) and non-flooded conditions in August 2014.**




**Table 1: Environmental characteristics of August 2014 for tide ratio study and days with spring and neap tides in May and October 2014, respectively.**

| Month | DOY | Tide type | Mean air temp (°C) | Mean soil temp (°C) | Mean PAR ($\mu mol\ m^{-2}\ s^{-1}$) | Mean plant height (m) |
|---|---|---|---|---|---|---|
| August* | - | - | 27.5 ± 2.6 | 29.1 ± 1.0 | 885 ± 606 | 0.61 ± 0.45 |
| May | 124-125 (-2)** | Neap | 21.40 ± 4.08 | 22.30 ± 1.30 | 1269 ± 599 | 0.64 ± 0.38 |
| | 137-138 (+3) | Spring | 18.74 ± 2.34 | 23.46 ± 0.84 | 1299 ± 573 | 0.64 ± 0.38 |
| Oct | 289-290 (+1) | Neap | 18.32 ± 3.42 | 21.74 ± 0.82 | 1108 ± 472 | 0.56 ± 0.41 |
| | 298-299 (+2) | Spring | 17.10 ± 3.50 | 20.26 ± 0.63 | 1059 ± 425 | 0.56 ± 0.41 |

Notes: *Based on monthly mean data; **No. in brackets represent day/s away from the actual neap and spring tide days. Negative and positive values indicate day/s before and after the actual neap and spring tide days, respectively.

**Table 2: $CO_2$ flux in exposed soil and submerged condition during night- and daytime on spring tide days for May and October 2014. (mean ± standard error).**

| | May | October |
|---|---|---|
| **Daytime** | | |
| Tidal Inundation | -4.90 ± 1.42 $\mu mol\ m^{-2}\ s^{-1}$ | -2.49 ± 0.65 $\mu mol\ m^{-2}\ s^{-1}$ |
| Exposed soil | -6.77 ± 1.11 $\mu mol\ m^{-2}\ s^{-1}$ | -3.59 ± 0.64 $\mu mol\ m^{-2}\ s^{-1}$ |
| **Nighttime** | | |
| Tidal Inundation | 1.37 ± 0.21 $\mu mol\ m^{-2}\ s^{-1}$ | 1.11 ± 0.63 $\mu mol\ m^{-2}\ s^{-1}$ |
| Exposed soil | 1.42 ± 0.02 $\mu mol\ m^{-2}\ s^{-1}$ | 1.26 ± 0.26 $\mu mol\ m^{-2}\ s^{-1}$ |

Notes: *Daytime data were from 0830 to 1630 hours.

**Table 3: Average of $CO_2$ fluxes on neap and spring tide days (mean ± standard error).**

| | May | October |
|---|---|---|
| Neap tide day | -1.92 ± 3.92 $\mu mol\ m^{-2}\ s^{-1}$ | -0.43 ± 2.89 $\mu mol\ m^{-2}\ s^{-1}$ |
| Spring tide day | -1.75 ± 3.43 $\mu mol\ m^{-2}\ s^{-1}$ | -0.35 ± 2.10 $\mu mol\ m^{-2}\ s^{-1}$ |

**Table 4: Daytime monthly reduction in August 2014 based on tide ratio group**

| Tide ratio | Frequency (%) | *$F_{mea}$ ($\mu mol\ m^{-2}\ s^{-1}$) | *$F_{mod}$ ($\mu mol\ m^{-2}\ s^{-1}$) | $CO_2$ reduction (%) |
|---|---|---|---|---|
| 0-0.5 | 75 | -680.33 | -751.30 | 9 |
| 0.5-0.75 | 15 | -88.61 | -126.17 | 30 |
| 0.75-1.0 | 10 | -48.36 | -80.63 | 40 |

Notes: *$F_{mea}$ is measured $CO_2$ flux; $F_{mod}$ is calculated $CO_2$ flux from a light response curve for $CO_2$ exchange model during non-flooded conditions.