# Peer review of "Influence of Tidal Inundation on CO2 Exchange between Salt Marshes and the Atmosphere"

_Biogeosciences, 2017_

## Referee Comment (RC1) · I. Forbrich (Referee) · 25 Sep 2017

I. Forbrich (Referee)

iforbrich@mbl.edu

The manuscript demonstrates the instantaneous effects of tidal inundation on daytime CO2 exchange measured using eddy covariance during one year in a salt marsh in Georgia, USA. The authors aim to determine the effect of inundation on CO2 exchange and quantify it. The focus of the study is on the reduction in daytime NEE, even though the authors state that they detect small reductions in nighttime fluxes as well. The authors use a ratio of vegetation height and tide height relative to the surface to classify how much of the canopy is submerged. To quantify the observed reduction in NEE during inundation, they model NEE during non-flooded situation by fitting a light response curve to daytime data when the marsh is not flooded. Subsequently, they use these modeled fluxes as a reference and calculate the difference between measured and modeled fluxes as a measure of flux reduction.

Eddy covariance measurements in tidal wetlands are still rare and new data offer valuable information about contemporary carbon cycling in these systems. However, even though there are not many studies, all of the published studies address the impact of tidal inundation. Thus, to me the most interesting aspect of this study is the approach used to quantify the reduction in CO2 fluxes, because this pattern seems to be consistent in salt marshes. However, I find the results mostly descriptive and not always consistent with the method description (see detailed comments below). Overall, I am missing a discussion of the advantages and/or disadvantages of this approach compared to earlier approaches (e.g. Kathilankal et al. (2008) or Forbrich & Giblin (2015)) and of implications for contemporary carbon cycling.

Major comments:
- The authors argue that with future sea level rise (and subsequent prolonged inundation), CO2 net uptake might be lower in the future and the marsh will convert into a mudflat. I disagree with this: Many studies have shown that – while biomass production is important – it is not the only driver for the long-term stability of salt marshes with regard to sea level rise. Mostly it will depend on interactions between factors such as biomass production, sediment availability, tide range, rate of sea level rise as well as the possibility to transgress further inland (e.g. Morris et al. 2002, Kirwan et al. 2010, Kirwan et al. 2016).

- My impression is that the description of the approach and the results are contradictory: From the results (Section 3.3, 3.4, Fig. 14, Tab. 4) I take it, that the August CO2 fluxes were grouped in three classes based on the tide ratio and a light response curve was fitted to them separately as well as to 'non-flooded' conditions. This is not how I understood the methods (Section 2.4): I expected the light response curve to be fitted only to the CO2 fluxes under non-flooded conditions (to get a reference value

for non-flooded conditions). Afterwards, the modelled fluxes would be subtracted from the observed fluxes - independently from the tide ratio – to quantify the flux reduction. In a revised version of the manuscript, this should be explained better. It is not clear to me why the light response curve is fitted to $CO_2$ fluxes measured during partially or completely submerged conditions. I thought, the data coverage is so good that you know the magnitude of the 'real' fluxes, but you need to estimate how large they were if there was no tidal flooding (thus the reference value). Subsequently, I am not sure how to interpret the values for *Fmea* and *Fmod* in Tab. 4. Especially since the time series is not continuous (since the night time data are not used), I think the only time period that give us reasonable information is each single daytime flooding event. Thus, I suggest that the difference between *Fmea* and *Fmod* (only determined for non-flooded conditions) be calculated for each single daytime tide event and grouped according to tide ratio afterwards.

- The comparison of neap and spring tide conditions in May and October is only descriptive and not connected to the fitting approach. I suggest to using the fitting approach for each month of the year and use these selected days to demonstrate the approach described above.

Minor comments:

page 1 ll. 22-25: Are all these numbers from the Chmura paper? Otherwise, they need references.
Page 2 l. 2 delete 'of'
page 2 ll 10-10-13: see comments above: There are biogeomorphic feedbacks between vegetation cover, tidal inundation and accretion rates, that are not directly linked to instantaneous $CO_2$ exchange but help marshes to keep their position relative to mean sea level.
Page 2 ll. 29: I would rephrase that, do you 'hypothesize' this rather than 'believe'?
Page 2 ll30: delete 'also'

Page 3 ll 6-9: Can you mention the height differences of the tall, medium and short plants? Which one do you use for the tide ratio? Also, how much variation is there during the entire growing season?
Page 3 ll12-14: You do not need to say here that tides affect $CO_2$ exchange greatly, just mention the tide range.
Page 3 ll23: Are the tide heights reported in NAVD88 or relative to surface?
Page 4 ll. 7: Which quality control steps were applied?
Page 4 ll18-25: See comments above
Page 4 ll26 – page 5 ll 5: Considering the high quality of the data set, I am surprised that you pick only one month and a couple of days to assess the tidal influence. The data coverage especially during the day is high and it would be possible to do this over the entire year and not only restrict yourself to the same climatic conditions (i.e. high irradiation).
Page 5 ll 14 and ll 20-21: Contrary to these statements, Fig 8 shows that the marsh surface IS flooded during spring tide?!
Page 6 ll 2-8: Most of this is descriptive and shown in Fig. 10 anyway. However, the observation that plants suffered from heat stress in July and August is interesting and would merit more analysis and discussion.
Page 6 ll 14 – Page 7 ll5: See comments above
Page 7 ll 7 – 11: See comments above
Page 7 ll12 – 18: Why do you compare two random days (September as opposed to May, October or August as previously used) to give an example for the flux reduction instead of describing the results from the fitting procedure?
Page 7 ll 21-24: I think this should go into 'results'.
Page 9 ll 2-5: See comments above: $CO_2$ exchange might be reduced instantaneously during inundation but that cannot be extrapolated over long periods of time.

Figures:
Fig. 1 and 4 are not really necessary.

Fig. 3 and Fig. 6 could be combined.
Fig. 11: This would work better with days that have been analyzed or discussed before (e.g. May/October).
Fig. 12 and 13 are not really necessary.
Fig. 14 needs more explanation: E.g., the different symbols are not explained, only the fit.

Table 1: All the values are given in the text, so this table is a repetition. Either change the text or remove the table.
Table 4 : See comments above.

References:
Forbrich & Giblin 2015, Journal of Geophysical Research
Kathilankal et al. 2008, Environmental Research Letters
Kirwan et al. 2010, Geophysical Research Letters
Kirwan et al. 2016, Nature Climate Change
Moffett et al. 2010, Water Resources Research
Morris et al. 2002, Ecology

---

## Referee Comment (RC2) · P. Polsenaere (Referee) · 16 Oct 2017

**GENERAL COMMENTS:**

The submitted manuscript of Nahrawi et al. under review for journal Biogeosciences presents the tidal rhythm effect on atmospheric Eddy Covariance (EC) CO2 fluxes between a salt marsh situated close to Sapelo Island (GA) and the atmosphere. Through three chosen data sets obtained in 2014, EC CO2 fluxes are described and compared at the daily and monthly scales (neap/spring tides) during different Spartina alterniflora air exposures according to vegetation amounts/biomass and tidal water levels. Salt marshes represent key coastal systems among carbon budgets where the highest carbon assimilation rates in the biosphere are measured. These heterogeneous and dynamic coastal systems remain particularly hard to study due to the high spatio-temporal heterogeneity in terms of CO2 fluxes at terrestrial-aquatic exchange interfaces. Then, this study is of particular interest.

- My first general concern is, although the authors obtained an interesting and original full-year EC dataset, the tidal effect on CO2 fluxes is solely addressed at the daily and monthly scales through small chosen data parts. Annual air-marsh CO2 fluxes could be presented and discussed as well, to clearly quantify the tidal influence on the carbon budget of the studied marsh at the seasonal and annual scales. It is too bad as the EC technique allows computing such annual CO2 exchanges through continuous and non-invasive measurements during particular periods (i.e. flooded and non-flooded). Although well quantified, the tidal effect on CO2 fluxes is only shown through three chosen periods in 2014 on purpose. To go further and gain a real interest for the scientific community working on carbon budget over coastal systems, the manuscript should present or at least discuss the significance of the tidal effect on air-marsh CO2 exchanges and associated partitioned metabolic fluxes (i.e. NPP, GPP and CR) at the annual scale in my opinion (please see for instance Rocha and Goulden, J. Geophys. Res., 113, 1-12, 2008 and cited references below).

- It leads to my second general concern on the submitted manuscript; I recognize that studies on carbon processes and fluxes over intertidal salt marshes are still scarce and their influence on adjacent water systems is maybe not the main point of the study here. However as the tidal rhythm influence is precisely addressed here, why the important "Marsh CO2 Pump" concept initially proposed by Wang and Cai (2004) at the same location and studied by others later (to conceptualize tidal marshes as atmospheric CO2 sink and inorganic carbon source to the coastal ocean) is not discussed here? The submitted manuscript as it stands now only deals with CO2 flux comparison during spring and neap tide periods without encompassing the annual scale for carbon budget computations. Studies dealing with carbon budget over similar coastal ecosystems

exist; the present study would significantly gain interest taking into account these latters and going toward the seasonal and annual scales as well. Please see studies of Guo et al. (Agr. Forest Meteorol., 149, 1820-1828, 2009), Yan et al. (Glob. Change Biol., 14, 1690-1702, 2008), Wang and Cai (Limnol. Oceanogr., 49, 341-354, 2004) and Wang et al. (Limnol. Oceanogr., 61, 1916-1931, 2016) for instance.

- My last general concern is authors clearly observed a CO2 flux reduction at high tide during the day in comparison with low tide periods as already observed over same coastal systems, i.e. salt marshes (Houghton and Woodwell, Ecology, 61, 1434-1445, 1980; Kathilankal et al., Env. Res. Lett., 3, 1-6, 2008) and elsewhere over intertidal flats (Zemmelink et al., Geophys. Res. Lett., 36, 2009; Polsenaere et al., Biogeosciences, 9, 249-268, 2012) or Amazon floodplain (Morison et al., Oecologia, 125, 400-411, 2000) for instance. However, no explanation is given or even discussed to try to understand mechanisms involved in this reduction, especially those taking place at the air-water or air-marsh interfaces or underwater through the different involved inorganic carbon forms (i.e. gas transfer velocity and water-air gas exchange, water pCO2 and DIC, GPP and CR as NEE drivers, ..., please see cited references and others). Please see the next comments among with cited references above to help in the revision of the different sections of the manuscript. I would recommend further revisions in this way to allow the publication of the present paper of Nahrawi et al. for the journal Biogeosciences.

**SPECIFIC COMMENTS:**

Abstract: - I.12, 14-15, 17-18: as the authors got a full-year EC CO2 flux dataset, analyzing the tidal effect on CO2 fluxes for each month of 2014 according to vegetation biomasses, tide ratio per month, etc...at the daily, seasonal and annual scales would give to the submitted manuscript much more consistency and interest (as explained above).

1 Introduction:

- In the introduction section, there is a shortage of references on the different studies

СЗ

dealing with carbon dynamics over salt marshes (air-marsh CO2 fluxes, lateral inorganic carbon fluxes/exports with adjacent systems...) but also over similar intertidal coastal systems (freshwater marsh, tidal flat, floodplain, ...) where tidal effects have also been studied not solely with EC technique (see Clavier et al., Aquatic Botany, 95, 24-30, 2011; Ouisse et al., Mar. Ecol. Prog. Ser., 437, 79-87, 2011 and others). No information/reference is given about the atmospheric EC technique too. Mechanisms involved in the control of CO2 fluxes over salt marshes are poorly explained (I.30-31). - There is also a lack of quantitative data from bibliography to indorse different statements (for instance I.3, 21-25). - With regards to objectives and as already explained, I would recommend to add explanations for the CO2 flux reduction during immersion in the two first objectives and add a main third objective integrating the seasonal and annual scales to go further toward carbon budgets of the studied salt marsh.

**2 Material and methods:**

2.1 Please remind studies that have already been carried out at the same place. 2.2 Lack of information: why were two EC systems deployed (nothing is explained in the whole manuscript)? Why was a 5m-height used for the EC sensors (see footprint calculations)? EC systems were deployed in July 2013 and only data from May, October and August 2014 are presented, why? The reader understands it is for Spring/Neap tides comparisons at different vegetation growths but nothing is explained about it; also between July 2013 and January 2014, what has happened? 2.3 Figure 5 justified the interest to use the whole data set of 2014. According to footprint calculations, could the authors give to the reader an estimation of the footprint size (5 meters high ok but what about surface roughness, wind speeds, turbulence etc. . .) and directions (two EC systems were used with two opposite directions)? What about the potential influence of water during measurements especially at low tide (neap tide)? According to tide periods, the footprint size is modified (varying sensor heights). 2.4 Please specify the non-linear model equation for Fmod. It is not clear for the reader as it stands in the submitted manuscript. The last paragraph I. 26-5 on "August 2014 data selection during clear sky only" needs to be better explained and justified. Same calculations done for each tide (Ftide), each month (Ftot), each season and finally over the whole year would be very instructive.

**3 Results:**

In all sections of this result part, no statistics are given to indorse CO2 flux or associated variable comparisons and correlations. Measured CO2 fluxes could be specified through NPP, GPP and CR values. The effect of immersion on these metabolic fluxes (instead of CO2 fluxes during the day and night only) could be studied to go further as mentioned before. Please see technical comments for comments on associated figures and tables. - I.26-27, p.5: I don't understand why a 0.4 tide ratio corresponds to 40% of submerged plant parts in August 2014? - The introductive paragraph in 3.2 sub-section is too general and imprecise and maybe useless as flux values are given next in 3.2.1 and 3.2.2 (I.2 "late morning to noon time"; I.7 "respiration rates ... increase…"; I.12 "…10 times…"?) - 3.2.2 I.26-28 " reduction", please quantify them! - 3.3 (and associated figure 14): interesting but are R2 significant? Here again, adding data from other months in 2014 (than August) will probably bring more consistency and significance to the analysis. - 3.4 I don't fully understand this sub-section at the end of the result part although the monthly analysis in August is interesting and should be done for other months (or seasons) over the year.

**4 Discussion:**

The discussion part needs to be reorganized and reviewed with regards to previous general comments. In the submitted manuscript, it rather corresponds to result (subsection 4.1 for instance) descriptions than a real discussion on carbon processes and fluxes over salt marshes with associated environmental controls. Very few references are cited. Again, I really believe orientating the paper toward carbon dynamics at both diurnal, seasonal and annual scales would deeply increase the impact of the paper to the scientific community working on such coastal systems. - I.1, p.8: " a net uptake

of CO2 during nighttime immersion": it is necessarily associated to inorganic carbon dynamics in water bodies close to the tidal marsh system (advection, hydrodynamic, air-water gas transfer velocity, ....). But it is not discuss in the submitted manuscript? - I.16, p.8: "a certain water table threshold"?; I.29-30: "140.79 micromol m-2 s-1 corresponding to as much as 15% of the total monthly reduction"? I don't understand the flux value; please review it.

**5. Conclusion:**

The first two paragraphs are too general and the third one should be specified with estimations of CO2 flux reduction by immersion at the annual scale from a carbon budget point of view. Also a point could be done here on the interest to use simultaneously the atmospheric EC and aquatic EC techniques (see Berg et al., Mar. Ecol. Prog. Ser., 261, 75-83, 2003 and other publications on Zostera marina seagrass meadows of the eastern shore of Virginia for more information on the technique) associated to water DIC measurements (cited references) to better measure and integrate salt marsh metabolism processes/fluxes during both emersion and immersion periods to specify the role of salt marshes among regional and global carbon budgets.

**TECHNICAL COMMENTS:**

- 14 figures are really too much.
- Figure 1 is maybe not necessary.

- Figure 3 (caption) needs to be specified to help the reader to understand exactly when the marsh is totally emerged, partially emerged/immersed and totally immersed during neap tides and spring tides. Spring and neap tides occur twice during each month so an associated table with number of hours during which the marsh is fully emerged/immersed and partially exposed to water during each month could be useful for instance. Fully exposed to air: low tides during neap tides? Low tides during spring tides? High tides during neap tides? Fully immersed: high tide during Spring tide?

High tide during neap tide? During transition periods (rising/ebbing) of the tide, how is the marsh? This part really needs to be clearer.

- Figure 4: as it is, this latter doesn't bring a lot of information (weak captions...)

- Figure 5: interesting figure that could be associated to a footprint analysis/diagram according to wind speeds/directions, surface roughness and turbulence. I am wondering why EC systems worked so poorly during nighttime periods (00:00-06:00 and 18:00-24:00). Could it only be explained by the associated low turbulence regimes at night!

- Figure 6: not very useful as this interesting parameter is comprehensible without figure.

- Figure 7: How many phenocam images were taken during the study? Four? The phenology of the salt marsh is particularly important in the control of carbon flux dynamic. Do satellite images exist for the studied area to estimate the relative contribution of the marsh in the EC footprint to measured CO2 fluxes?

- Figure 8: not necessary (cf. see previous comments).

- Figure 9: very informative figure except it concerns August 2014 only!

- Figure 10: very informative figure. Partitioned fluxes, i.e. NPP, GPP and CR could be specified on it.

- Figures 11,12 and 13: It is very hard for the reader to follow and understand these figures and associated text in the result sections (daily fluxes, mean fluxes?). All of them are not necessary. Wouldn't be possible to do one figure with averaged CO2 fluxes with corresponding immersed and emerged periods during day and night for each month? Keep Time of the day (24 hours) in x-axis and for the y-axis, add monthly mean CO2 fluxes (with associated SD), PAR and water level curves with clear/shaded areas for day/night periods respectively and vertical dotted lines for immersed/emerged periods for instance.

- Figure 14: see previous comments.

- Tables: see previous comments too.

---

## Referee Comment (RC3) · K. Moffett (Referee) · 18 Oct 2017

**Summary**

This manuscript presents monthly aggregates of a year of eddy covariance data from an intertidal salt marsh. From within that data, subsets are extracted for assessment of the reduction in net marsh CO2 uptake (Net Ecosystem Exchange, NEE) during daytime flooding tides (occurring during spring tide at this location) compared to days without mid-day flooding tides (occurring during neap tide at this location). A nondimensionalization of the effect of flooding tides on NEE is attempted via the ratio of tidal depth to vegetation height on the marsh platform. Some Spartina alterniflora

photosynthesis light response curves are also presented with some possible, but unclear, relation to tidal flooding (as methods are not disclosed). These three foci, NEE data, tide-to-vegetation ratio, and light response curves could be better integrated into a coherent through-line for the manuscript. Overall, although the manuscript presents interesting-looking data, I unfortunately cannot recommend it for publication. My criteria for recommending rejection are the lack of disclosure of multiple important aspects of the methodology, confusing presentation of the results, strikingly lacking review or thorough discussion of highly pertinent literature, and lack of substantively new scientific contribution beyond nearly replicating figures of prior papers with new data. In my opinion, revising the manuscript so as to fix these issues, especially the last one, would thereby generate a wholly new manuscript that should be considered afresh, not as a revision.

**Major Comments**

A. The overall premise of the study – that "documentation on the exchange of CO2 between salt marsh ecosystem and atmosphere measured by modern eddy-covariance systems are still very limited (Kathilankal et al., 2008)." (pg 2 line 17-18) – is substantially flawed in that the manuscript does not thoroughly review and cite literature review on this very specific topic. Specifically, 5 key progenitors to this manuscript are: Kathilankal et al., 2008; Moffett et al. 2010; Schafer et al. 2014; Artigas et al. 2015; Forbrich and Giblin, 2015.

These may not be all the relevant papers, but each of them has measured, analyzed, discussed, and published on the topic (tidal flooding effects on NEE) of this manuscript. Thoroughly reviewing these and other potentially related papers should have been the first responsibility executed by the study. In particular, there is no important physical difference between the "spring vs neap" factor that is the focus of this manuscript and the presence vs absence of tidal flooding studied by both Kathilankal et al., 2008 and Moffett et al. 2010.

It was not initially clear in this manuscript that the study would compare flooded to non-flooded conditions. This was suggested, but not clear, on page 3 line 13 "During high spring tide, most of the vegetation is submerged and exposed during low spring tide and neap tide period." Only upon getting to Figure 8 was it clear to this reader that the "neap tide" conditions actually represent "no flooding" from the perspective of the vegetation, so the comparison is flood vs no flood (not higher spring vs lower neap flood depth as this reader mistakenly assumed at first). If multiple prior studies have compared salt marsh NEE during flooded and non-flooded conditions, and even taken into account the effects of different flood depths (starting with Moffett et al. 2010), then what is the unique contribution intended by this manuscript?

B. The specific model used to calculate the CO2 exchange during non-flooded periods is not specified in the methods. All that is said is "Fmod is calculated CO2 flux from a light response curve for CO2 exchange model during non-flooded conditions," (page 4 line 24) with no model or methodological citation.

C. The methods paragraph beginning on page 4 with "Data of August 2014 was used to study..." is very unclear. After reading it 4 times and also referring to the table and figures I still cannot understand what analysis was done on the August data, what on the May, what on the October, and why the same analysis seems not to have been done on either all or just one of those time periods.

D. I am further concerned with the aspect of the study based on a tide-to-vegetation ratio. On page 4 the ratio was defined as (tide height) / (mean plant height). However, on page 5 and in Table 1, I see that the ranges of plant heights were quite large. It was reported on page 5:

- "The mean plant height in May 2014 was 0.64  $\pm$  0.38 m."

- "In October. . . the mean plant height was 0.56  $\pm$  0.41 m."

- "in August. . . monthly mean plant height was recorded at 0.61  $\pm$  0.45 m."

СЗ

It is not stated what the second number in these cases was (0.38, 0.41, and 0.45), but I assume it may be a standard deviation; if so, these results seem to say that the distribution of plant heights was very broad, with many plants of nearly zero height and also many of around a meter or more. If instead these second numbers are standard errors (as perhaps they should be?) then it suggests that the means are not at all well constrained. In either case, how then is the ratio (tide height) / (mean plant height) a metric that captures flood-vegetation interactions in a comprehensive way?

Lastly, the methods section did not include information on how plant height was surveyed, over what area, whether by plot sampling and extrapolation or some kind of exhaustive sampling, whether by LIDAR (which is impossible to use to obtain plant height and difficult to use even for sediment height over low-relief, low/soft vegetation marshes), etc., so it is impossible to interpret what these standard deviations or errors may be representing in terms of sampled variability.

E. Although Figure 14 appears interesting, I find it unpublishable as is since there was no disclosure in the Methods section of how these light response curves were obtained.

I am doubly concerned because I myself attempted some years ago (unpublished) using a LICOR 6400to gather light response curves from Spartina foliosa contained in a bucket in a laboratory and flooded to different depths. Over short terms – if using the rapid measurement technique of collecting data over only seconds to minutes at each flooding or light level – I did see what appeared to be response curves. However, I also conducted the study using the slow equilibration technique, collecting data for tens of minutes to hours for each flood or light level; those curves appeared bizarre and even inverse from what one would expect. Only after plotting all the data chronologically I realized I had actually measured the diurnal circadian cycle of the Spartina (due to the long day/evening in the lab of continuous experiments) and therefore negligible, if any, actual response to the flooding itself.

Although it is nearly certain that the authors conducted a more nuanced and thorough

experiment than my one failed attempt at such a thing, lacking any information about how the light response curve portions of the study were done, I cannot say! Likewise, I cannot have confidence in the conclusions of Section 3.3 without further methodological information.

Minor Comments

1. pg 1 line 8 – It is not appropriate to quote in the abstract a quantitative value, the precise magnitude of which is the subject of a whole field of ongoing research (this manuscript included), and for which other values have been offered (e.g., in Forbrich & Giblin 2015), especially when it is a value that was not derived by the study itself and is deprived of a proper citation (to Chmura et al 2003). \*Remove\* this value of 210 g C /m2 / yr from the abstract. Use a qualitative magnitude instead, if need be to make the point.

2. pg 1 line 13 – Amend to "The conditions with a high tide-to-vegetation height ratio..." Without reference to HEIGHT it is unclear what values are being divided. Look for this omission and correct throughout manuscript.

3. pg 1 line 14 – Amend to "...conditions with a low ratio." It is no more a "tide ratio" than it is a "vegetation ratio" – the numerator nor denominator can stand on its own, so just call it a ratio. Look for this confusion and correct throughout manuscript.

4. Figure 1 is not needed.

5. What are the sources of the ecoregion and land classification data in Figure 2? Should be cited.

6. Figure 3 not needed.

7. Figure 4 not needed.

8. Figure 5 seems to show that hardly any nighttime data were retained after QA/QC. Analysis and discussion should be provided of whether sufficient data remained to

make calculations and inferences at night. The figure should be moved to an appendix/supplement, however.

9. page 4 line 8 – I do not understand "Data from north and south systems were combined and selected based on the climatological footprint". Please explain further.

10. page 4 line 9 – I do not understand "Only measurements that contributed to more than 70% of the CO2 flux within the study area were used". Please explain further.

11. Figure 6 not needed.

12. Figure 7 not needed.

13. Figure 9 is not needed; also see Major Comments C and D, above, regarding related confusion as to what the study actually did.

14. Figure 10 is impressive and demonstrates the incredible volume of interesting data collected by the study team. However, see Minor Comment number 8 – I wonder a bit at the small standard deviations reported for nighttime NEE values given that the sample size after QA/QC was quite small for night times. The plot is very similar to that by Kathilankal et al., 2008 that spanned May through October, although this manuscript helpfully expands the figure through all 12 months.

15. Figure 11 appears nearly identical to the kind of data presented in Kathilankal et al., 2008 and in Moffett et al. 2010. What is the new scientific insight added by this study that warrants re-publishing a known phenomenon?

16. Sections 3.2.1 and 3.2.2 – The manuscript to this point has not made it clear to me why we should be interested to compare May and October data, and so I do not see the point of these sections or Figure 12 or 13. Recommend omitting.

17. Page 8 line 18-19. This manuscript writes "Site studies of these authors are dominated by marsh grass species which grow upright, either Spartina alterniflora (Kathilankal et al., 2008) or Spartina foliosa and Distichlis spicata (Forbrich and Giblin, 2015; Moffett et al., 2010)." This is a direct quote – actually a mis-quote – of Forbrich and Giblin 2015, who wrote (page 1835) "Sites studied by these authors are both dominated by marsh grass species which grow upright, either Spartina alterniflora [Kathilankal et al., 2008] or Spartina foliosa and Distichlis spicata [Moffett et al., 2010]." but also clarified that "At our site, Spartina patens often lies prostrate forming a dense, green carpet..." (hence the mis-quote). [And actually the site by Moffett et al. was as much Salicornia virginica as Spartina and Distichlis; west-coast US marshes are odd compared to east.]

18. If use of a digital online supplement is enabled by the journal, the figures to be removed could be provided in a supplement.

**References**

- Artigas, F., Shin, J. Y., Hobble, C., Marti-Donati, A., Schäfer, K. V. R., & Pechmann, I. (2015). Long term carbon storage potential and CO2 sink strength of a restored salt marsh in New Jersey. Agricultural and Forest Meteorology, 200, 313–321. https://doi.org/10.1016/j.agrformet.2014.09.012

- Forbrich, I., & Giblin, A. E. (2015). Marsh-atmosphere CO2 exchange in a New England salt marsh. Journal of Geophysical Research: Biogeosciences, 120(9), 1825–1838. https://doi.org/10.1002/2015JG003044

- Kathilankal, J. C., Mozdzer, T. J., Fuentes, J. D., D'Odorico, P., McGlathery, K. J., & Zieman, J. C. (2008). Tidal influences on carbon assimilation by a salt marsh. Environmental Research Letters, 3(4), 044010. https://doi.org/10.1088/1748-9326/3/4/044010

- Moffett, K. B., Wolf, A., Berry, J. A., & Gorelick, S. M. (2010). Salt marshatmosphere exchange of energy, water vapor, and carbon dioxide: Effects of tidal flooding and biophysical controls. Water Resources Research, 46(10), W10525. https://doi.org/10.1029/2009WR009041

- Schäfer, K. V. R., Tripathee, R., Artigas, F., Morin, T. H., & Bohrer, G. (2014). Carbon

dioxide fluxes of an urban tidal marsh in the Hudson-Raritan estuary: Carbon dioxide fluxes of an wetland. Journal of Geophysical Research: Biogeosciences, 119(11), 2065–2081. https://doi.org/10.1002/2014JG002703

---

## Author Comment (AC1) · 11 Dec 2017

**Response to Referee Comment 2 (RC2)**

**GENERAL COMMENTS:**

**Comment 1:** My first general concern is, although the authors obtained an interesting and original full-year EC dataset, the tidal effect on $CO_2$ fluxes is solely addressed at the daily and monthly scales through small chosen data parts. Annual air-marsh $CO_2$ fluxes could be presented and discussed as well, to clearly quantify the tidal influence on the carbon budget of the studied marsh at the seasonal and annual scales. It is too bad as the EC technique allows computing such annual $CO_2$ exchanges through continuous and non-invasive measurements during particular periods (i.e. flooded and non-flooded). Although well quantified, the tidal effect on $CO_2$ fluxes is only shown through three chosen periods in 2014 on purpose. To go further and gain a real interest for the scientific community working on carbon budget over coastal systems, the manuscript should present or at least discuss the significance of the tidal effect on air-marsh $CO_2$ exchanges and associated partitioned metabolic fluxes (i.e. NPP, GPP and CR) at the annual scale in my opinion (please see for instance Rocha and Goulden, J. Geophys. Res., 113, 1-12, 2008 and cited references below).

**Response 1:** We will provide more data instead of just using one month of data (August) to better explain the total monthly flux reduction based on our approach. For carbon budget, we have separate paper that discuss on that matter. This submitted manuscript only focused on tidal effect on $CO_2$ fluxes and how this reduction is translated into quantitative estimation of total monthly reduction for the study month.

**Comment 2:** It leads to my second general concern on the submitted manuscript; I recognize that studies on carbon processes and fluxes over intertidal salt marshes are still scarce and their influence on adjacent water systems is maybe not the main point of the study here. However, as the tidal rhythm influence is precisely addressed here, why the important "Marsh $CO_2$ Pump" concept initially proposed by Wang and Cai (2004) at the same location and studied by others later (to conceptualize tidal marshes as atmospheric $CO_2$ sink and inorganic carbon source to the coastal ocean) is not discussed here? The submitted manuscript as it stands now only deals with CO2 flux comparison during spring and neap tide periods without encompassing the annual scale for carbon budget computations. Studies dealing with carbon budget over similar coastal ecosystems exist; the present study would significantly gain interest taking into account these latters and going toward the seasonal and annual scales as well. Please see studies of Guo et al. (Agr. Forest Meteorol., 149, 1820-1828, 2009), Yan et al. (Glob. Change Biol., 14, 1690-1702, 2008), Wang and Cai (Limnol. Oceanogr., 49, 341-354, 2004) and Wang et al. (Limnol. Oceanogr., 61, 1916-1931, 2016) for instance.

**Response 2:** We will look into the suggested references. However, one of the main purpose of this study is to quantify the daytime monthly reduction of $CO_2$ fluxes. Most of the study only reported the instantaneous flux reduction rather than looking at a longer interval such as monthly basis. By

having some knowledge on how much monthly (or even seasonally and annually) reduction in $CO_2$ fluxes, we will have a bigger and better picture on how much overall reduction for some interval of time which are more meaningful than just looking at a single point of time which most reported by previous study.

**Comment 3:** My last general concern is authors clearly observed a $CO_2$ flux reduction at high tide during the day in comparison with low tide periods as already observed over same coastal systems, i.e. salt marshes (Houghton and Woodwell, Ecology, 61, 1434-1445, 1980; Kathilankal et al., Env. Res. Lett., 3, 1-6, 2008) and elsewhere over intertidal flats (Zemmelink et al., Geophys. Res. Lett., 36, 2009; Polsenaere et al., Biogeosciences, 9, 249-268, 2012) or Amazon floodplain (Morison et al., Oecologia, 125, 400-411, 2000) for instance. However, no explanation is given or even discussed to try to understand mechanisms involved in this reduction, especially those taking place at the air-water or air-marsh interfaces or underwater through the different involved inorganic carbon forms (i.e. gas transfer velocity and water-air gas exchange, water $pCO_2$ and DIC, GPP and CR as NEE drivers, . . ., please see cited references and others). Please see the next comments among with cited references above to help in the revision of the different sections of the manuscript. I would recommend further revisions in this way to allow the publication of the present paper of Nahrawi et al. for the journal Biogeosciences.

**Response 3:** Thank you for pointing this out. We will look in details on this subject and incorporate it in the revised version of the manuscript. However, as mentioned earlier, since most of previous studies only reported the instantaneous reduction of $CO_2$ flux, we, in this paper, use monthly basis to quantify $CO_2$ reduction in salt marsh ecosystem due to tidal inundation. The reduction could be small instantaneously but very significant when we quantify it for a long-term period (i.e monthly, seasonally or annually) which gives more meaning than just at one single point of time.

**SPECIFIC COMMENTS:**

**Comment – Abstract:** - l.12, 14-15, 17-18: as the authors got a full-year EC $CO_2$ flux dataset, analyzing the tidal effect on CO2 fluxes for each month of 2014 according to vegetation biomasses, tide ratio per month, etc. .at the daily, seasonal and annual scales would give to the submitted manuscript much more consistency and interest (as explained above).

**Response – Abstract:** Yes, we have full-year EC $CO_2$ flux dataset. However, in this paper, we only focus on effect on tidal on $CO_2$ reduction and to emphasis how we quantify the monthly basis of $CO_2$ reduction. Besides, there is still lack of studies on neap and spring tide comparison in terms of $CO_2$ reduction.

1 Introduction:

**Comment 1.1**: In the introduction section, there is a shortage of references on the different studies dealing with carbon dynamics over salt marshes (air-marsh $CO_2$ fluxes, lateral inorganic carbon fluxes/exports with adjacent systems. . .) but also over similar intertidal coastal systems (freshwater marsh, tidal flat, floodplain, . . .) where tidal effects have also been studied not solely with EC technique (see Clavier et al., Aquatic Botany, 95, 24-30, 2011; Ouisse et al., Mar. Ecol. Prog. Ser., 437, 79-87, 2011 and others). No information/reference is given about the atmospheric EC technique too. Mechanisms involved in the control of $CO_2$ fluxes over salt marshes are poorly explained (l.30-31).

**Response 1.1:** We will include more information on the stated subject based on the existing references in the revised version of the manuscript. Thank you for pointing out several references related to the subject. We will also add more information and references related to EC technique in the revised version of the manuscript.

**Comment 1.2:** There is also a lack of quantitative data from bibliography to indorse different statements (for instance l.3, 21-25). - With regards to objectives and as already explained, I would recommend to add explanations for the $CO_2$ flux reduction during immersion in the two first objectives and add a main third objective integrating the seasonal and annual scales to go further toward carbon budgets of the studied salt marsh.

**Response 1.2:** We will add more explanations for the $CO_2$ flux reduction during immersion as suggested. However, as mentioned earlier, this paper only focuses on the effect of tide on $CO_2$ exchange and we are in the final stage of preparing our paper on carbon budget in similar ecosystem.

2 Material and methods:

**Comment 2.1:** Please remind studies that have already been carried out at the same place.

**Response 2.1:** Thank you for the reminder and we will look into it.

**Comment 2.2:** Lack of information: why were two EC systems deployed (nothing is explained in the whole manuscript)? Why was a 5m-height used for the EC sensors (see footprint calculations)? EC systems were deployed in July 2013 and only data from May, October and August 2014 are presented, why? The reader understands it is for Spring/Neap tides comparisons at different vegetation growths but nothing is explained about it; also between July 2013 and January 2014, what has happened?

**Response 2.2:** The complexity and heterogeneity of the ecosystems drove us to use two EC systems at the study area. Installing two EC systems at one flux tower minimise the gaps in the data due to maintenance and calibration, instruments malfunction, and accommodates seasonal

changes in changing in wind direction. The south system facing south covers the angle from 90° to 270° and the rest of the area is covered by the north system facing north. The instrumentation is 5 m above ground, assuming that footprint sampled by the tower has a radius ranging from 0.5 to 1 km. However, it still depending on the wind direction and fetch, surface roughness, measurement height and atmospheric stability. Based on the footprint calculation, we make sure that only areas that are covered with *Spartina alterniflora* is being sampled. As mentioned in section 2.4 page 4 l. 26 – 26 and page 5 l. 1 – 5, only days with clear sky condition during spring tide and neap tide days were used. There were very limited days with such condition. Therefore, we only able to use days in May and October for neap and spring tide comparisons in our study. The two different months represent the comparison between neap and spring tide days. Meanwhile, August data was used randomly to quantitatively estimate monthly $CO_2$ reduction. These are the two main objectives in our study. We only select the data based on specific cases between July 2013 and January 2014.

**Comment 2.3:** Figure 5 justified the interest to use the whole data set of 2014. According to footprint calculations, could the authors give to the reader an estimation of the footprint size (5 meters high ok but what about surface roughness, wind speeds, turbulence etc.) and directions (two EC systems were used with two opposite directions)? What about the potential influence of water during measurements especially at low tide (neap tide)? According to tide periods, the footprint size is modified (varying sensor heights).

**Response 2.3:** Based on our footprint calculation, the footprint is approximately from 300m and can be up to 8000m. We will include some ideas on the component mentioned above in our revised manuscript as well as wind direction. We did not see potential influence of water during neap tide. The sensors height was not modified throughout the study period.

**Comment 2.4:** Please specify the non-linear model equation for Fmod. It is not clear for the reader as it stands in the submitted manuscript. The last paragraph l. 26-5 on "August 2014 data selection during clear sky only" needs to be better explained and justified. Same calculations done for each tide (Ftide), each month (Ftot), each season and finally over the whole year would be very instructive.

**Response 2.4:** We used August 2014 daytime data to estimate monthly $CO_2$ reduction. For neap and spring tide comparison, only days with clear sky were selected. We will specify in detail the equation and calculations for Fmod, Ftide and Ftot in the revised version of the manuscript.

3 Results:

**Comment 3.1:** In all sections of this result part, no statistics are given to indorse $CO_2$ flux or associated variable comparisons and correlations.

**Response 3.1:** We will include a statistical analysis in the revised version of the manuscript.

**Comment 3.2:** Measured $CO_2$ fluxes could be specified through NPP, GPP and CR values. The effect of immersion on these metabolic fluxes (instead of $CO_2$ fluxes during the day and night only) could be studied to go further as mentioned before.

**Response 3.2:** In this paper, we are only interested in the effect of tide on $CO_2$ exchange. We might not be able to include the subject in this paper because we are in the final stage preparing a paper related to the subject.

**Comment 3.3:** Please see technical comments for comments on associated figures and tables. - l.26-27, p.5: I don't understand why a 0.4 tide ratio corresponds to 40% of submerged plant parts in August 2014? –

**Response 3.3:** The tide ratio was calculated as: Tide ratio = Zt/h

Where Zt is the tide height and h is the mean plant height. When the tide ratio is equals to 1 (or 100%, if it is converted into percentage), the plants are completely submerged and the plant were completely exposed to the atmosphere when the tide ratio is equals to 0. Therefore, we assumed that based on the tide ratio, we converted the value into percentage which represents how much the plant is submerged.

**Comment 3.4:** The introductive paragraph in 3.2 sub-section is too general and imprecise and maybe useless as flux values are given next in 3.2.1 and 3.2.2 (l.2 "late morning to noon time"; l.7 "respiration rates . . . increase. . ."; l.12 ". . .10 times. . ." ?) - 3.2.2 l.26-28 " reduction", please quantify them!

**Response 3.4:** We will quantify them as suggested.

**Comment 3.5:** 3.3 (and associated figure 14): interesting but are R2 significant? Here again, adding data from other months in 2014 (than August) will probably bring more consistency and significance to the analysis.

**Response 3.5:** We will include statistical analysis in the revised version of the manuscript. We will also add more months in the study as suggested.

**Comment 3.6:** 3.4 I don't fully understand this sub-section at the end of the result part although the monthly analysis in August is interesting and should be done for other months (or seasons) over the year.

**Response 3.6:** We will explain in detail how we obtained the data in this section in the revised version of the manuscript. We will also add more months (minimum 4 months that represents different seasons) for comparison.

4 Discussion:

**Comment 4.1:** The discussion part needs to be reorganized and reviewed with regards to previous general comments. In the submitted manuscript, it rather corresponds to result (subsection 4.1 for instance) descriptions than a real discussion on carbon processes and fluxes over salt marshes with associated environmental controls. Very few references are cited. Again, I really believe orientating the paper toward carbon dynamics at both diurnal, seasonal and annual scales would deeply increase the impact of the paper to the scientific community working on such coastal systems.

**Response 4.1:** We will reorganize and review our paper as suggested. We will also cite more reference to strengthen our findings on the subject matter in the revised version of the manuscript.

**Comment 4.2:** l.1, p.8: " a net uptake of $CO_2$ during nighttime immersion": it is necessarily associated to inorganic carbon dynamics in water bodies close to the tidal marsh system (advection, hydrodynamic, air-water gas transfer velocity, . . ..). But it is not discuss in the submitted manuscript?

**Response 4.2:** We admit that we did In this study, we did not discuss much night time fluxes as how we described the day time fluxes. It is because, we lost a lot of night time data due to a very low u* that resulted in a very large footprint. Besides, we have lack of information related to nighttime condition of the study area, thus we focused more on daytime.

**Comment 4.3:** l.16, p.8: "a certain water table threshold"?; l.29-30: "140.79 micromol m-2 s-1 corresponding to as much as 15% of the total monthly reduction"? I don't understand the flux value; please review it.

**Response 4.3:** We will review it and provide details explanation on how we obtained the data in the revised version of the manuscript. Some previous study had introduced a water table threshold (Kathilankal et. al, 2008; Forbrich and Giblin, 2015). The mentioned value was the overall daytime $CO_2$ reduction in August. Detail explanation will be included in the revised manuscript.

5. Conclusion:

**Comment:** The first two paragraphs are too general and the third one should be specified with estimations of $CO_2$ flux reduction by immersion at the annual scale from a carbon budget point of view. Also a point could be done here on the interest to use simultaneously the atmospheric EC

and aquatic EC techniques (see Berg et al., Mar. Ecol. Prog. Ser., 261, 75-83, 2003 and other publications on Zostera marina seagrass meadows of the eastern shore of Virginia for more information on the technique) associated to water DIC measurements (cited references) to better measure and integrate salt marsh metabolism processes/fluxes during both emersion and immersion periods to specify the role of salt marshes among regional and global carbon budgets.

**Response:** We will revise our conclusion based on the suggestion. Thank you for pointing many important things which give us more ideas how to improve our manuscript.

**TECHNICAL COMMENTS:**

**Comment 1:** 14 figures are really too much.

**Response 1:** We will try to revise this figure so that it won't look too crowded and easy to digest.

**Comment 2:** Figure 1 is maybe not necessary.

**Response 2:** Figure 1 will be deleted

**Comment 3:** Figure 3 (caption) needs to be specified to help the reader to understand exactly when the marsh is totally emerged, partially emerged/immersed and totally immersed during neap tides and spring tides. Spring and neap tides occur twice during each month so an associated table with number of hours during which the marsh is fully emerged/immersed and partially exposed to water during each month could be useful for instance. Fully exposed to air: low tides during neap tides? Low tides during spring tides? High tides during neap tides? Fully immersed: high tide during Spring tide?

**Response 3:** We also believe it is a good idea to include all the mentioned information as suggested above. We will try to provide all the information in the revised manuscript.

---

## Author Comment (AC2) · 11 Dec 2017

**Response to Referee Comment 3 (RC3)**

**Major Comments:**

**Comment A:** The overall premise of the study – that "documentation on the exchange of $CO_2$ between salt marsh ecosystem and atmosphere measured by modern eddy-covariance systems are still very limited (Kathilankal et al., 2008)." (pg 2 line 17-18) – is substantially flawed in that the manuscript does not thoroughly review and cite literature review on this very specific topic. Specifically, 5 key progenitors to this manuscript are: Kathilankal et al., 2008; Moffett et al. 2010; Schafer et al. 2014; Artigas et al. 2015; Forbrich and Giblin, 2015. These may not be all the relevant papers, but each of them has measured, analyzed, discussed, and published on the topic (tidal flooding effects on NEE) of this manuscript. Thoroughly reviewing these and other potentially related papers should have been the first responsibility executed by the study. In particular, there is no important physical difference between the "spring vs neap" factor that is the focus of this manuscript and the presence vs absence of tidal flooding studied by both Kathilankal et al., 2008 and Moffett et al. 2010. It was not initially clear in this manuscript that the study would compare flooded to non-flooded conditions. This was suggested, but not clear, on page 3 line 13 "During high spring tide, most of the vegetation is submerged and exposed during low spring tide and neap tide period." Only upon getting to Figure 8 was it clear to this reader that the "neap tide" conditions actually represent "no flooding" from the perspective of the vegetation, so the comparison is flood vs no flood (not higher spring vs lower neap flood depth as this reader mistakenly assumed at first). If multiple prior studies have compared salt marsh NEE during flooded and non-flooded conditions, and even taken into account the effects of different flood depths (starting with Moffett et al. 2010), the what is the unique contribution intended by this manuscript?

**Response A:** Thank you for your comment. We are aware that there are similar studies about the effect of tide on $CO_2$ exchange. However, these studies mostly reported the instantaneous $CO_2$ reduction due to tide and mostly descriptively compare flood with no flood. In our study, we try to quantitatively estimate the reduction of $CO_2$ and expand our estimation to monthly basis. We believe that, by having a quantitatively data on the reduction, we would have a clearer picture on how much the reduction we are talking about.

**Comment B:** The specific model used to calculate the $CO_2$ exchange during non-flooded periods is not specified in the methods. All that is said is "Fmod is calculated $CO_2$ flux from a light response curve for $CO_2$ exchange model during non-flooded conditions," (page 4 line 24) with no model or methodological citation.

**Response B:** We will specify in detail the model that we used to calculate $CO_2$ exchange during non-flooded periods in the revised version of the manuscript.

**Comment C:** The methods paragraph beginning on page 4 with "Data of August 2014 was used to study. . ." is very unclear. After reading it 4 times and also referring to the table and figures I still cannot understand what analysis was done on the August data, what on the May, what on the

October, and why the same analysis seems not to have been done on either all or just one of those time periods.

**Response C:** August data was used to quantitatively estimate daytime monthly $CO_2$ reduction. This month was selected randomly to demonstrate the daytime monthly reduction of $CO_2$. We will add more months (different season) to put more meaning into this estimation. As mentioned in section 2.4 page 4 line 26 – 28 and page 5 line 1 – 5, only days with clear sky condition during spring tide and neap tide days were used. There were very limited days with such condition. Therefore, we only able to use days in May and October for neap and spring tide comparisons in our study. The two different months represent the comparison between neap and spring tide days.

**Comment D:** I am further concerned with the aspect of the study based on a tide-to-vegetation ratio. On page 4 the ratio was defined as (tide height) / (mean plant height). However, on page 5 and in Table 1, I see that the ranges of plant heights were quite large. It was reported on page 5:

- "The mean plant height in May 2014 was $0.64 \pm 0.38$ m."

- "In October. . . the mean plant height was $0.56 \pm 0.41$ m."

- "in August. . . monthly mean plant height was recorded at $0.61 \pm 0.45$ m."

It is not stated what the second number in these cases was (0.38, 0.41, and 0.45), but I assume it may be a standard deviation; if so, these results seem to say that the distribution of plant height was very broad, with many plants of nearly zero height and also many of around a meter or more. If instead these second numbers are standard errors (as perhaps they should be?) then it suggests that the means are not at all well constrained. In either case, how then is the ratio (tide height) / (mean plant height) a metric that captures flood-vegetation interactions in a comprehensive way? Lastly, the methods section did not include information on how plant height was surveyed, over what area, whether by plot sampling and extrapolation or some kind of exhaustive sampling, whether by LIDAR (which is impossible to use to obtain plant height and difficult to use even for sediment height over low-relief, low/soft vegetation marshes), etc., so it is impossible to interpret what these standard deviations or errors may be representing in terms of sampled variability.

**Response D:** Spartina has a very wide range of plant height and classify as three different types; short, medium and tall. We used mean value to calculate our tide ratio. A quadrat survey was conducted for plant sampling and the plant height was measured monthly. The second number represents standard deviation (not standard error as in manuscript, apologize for the error). We are aware that we simplified the method by using the average of the plant height. However, we believe that this potential method could be improved in the future and be used as a tool to estimate the amount of biomass exposed to the atmosphere.

**Comment E:** Although Figure 14 appears interesting, I find it unpublishable as is since there was no disclosure in the Methods section of how these light response curves were obtained. I am doubly concerned because I myself attempted some years ago (unpublished) using a LICOR 6400 to gather light response curves from Spartina foliosa contained in a bucket in a laboratory and flooded to different depths. Over short terms – if using the rapid measurement technique of collecting data over only seconds to minutes at each flooding or light level – I did see what appeared to be response curves. However, I also conducted the study using the slow equilibration technique, collecting data

for tens of minutes to hours for each flood or light level; those curves appeared bizarre and even inverse from what one would expect. Only after plotting all the data chronologically I realized I had actually measured the diurnal circadian cycle of the Spartina (due to the long day/evening in the lab of continuous experiments) and therefore negligible, if any, actual response to the flooding itself. Although it is nearly certain that the authors conducted a more nuanced and thorough experiment than my one failed attempt at such a thing, lacking any information about how the light response curve portions of the study were done, I cannot say! Likewise, I cannot have confidence in the conclusions of Section 3.3 without further methodological information.

**Response E:** In this study, we used direct relationship between $CO_2$ fluxes and PAR to obtain the light response curve. We will explain in detail the method that lead to Figure 14 in the revised version of the manuscript.

**Minor Comments:**

**Comment 1:** pg 1 line 8 – It is not appropriate to quote in the abstract a quantitative value, the precise magnitude of which is the subject of a whole field of ongoing research (this manuscript included), and for which other values have been offered (e.g., in Forbrich & Giblin 2015), especially when it is a value that was not derived by the study itself and is deprived of a proper citation (to Chmura et al 2003). *Remove* this value of 210 g C /m2 / yr from the abstract. Use a qualitative magnitude instead, if need be to make the point.

**Response 1:** The value has been removed.

**Comment 2:** pg 1 line 13 – Amend to "The conditions with a high tide-to-vegetation height ratio. . ." Without reference to HEIGHT it is unclear what values are being divided. Look for this omission and correct throughout manuscript.

**Response 2:** Corrected

**Comment 3:** pg 1 line 14 – Amend to ". . .conditions with a low ratio." It is no more a "tide ratio" than it is a "vegetation ratio" – the numerator nor denominator can stand on its own, so just call it a ratio. Look for this confusion and correct throughout manuscript.

**Response 3:** Corrected

**Comment 4:** Figure 1 is not needed.

**Response 4:** Removed from the manuscript

**Comment 5:** What are the sources of the ecoregion and land classification data in Figure 2? Should be cited.

**Response 5:** Will add the citation.

**Comment 6:** Figure 3 not needed.

**Response 6:** Noted

**Comment 7:** Figure 4 not needed.

**Response 7:** Noted

**Comment 8:** Figure 5 seems to show that hardly any nighttime data were retained after QA/QC.

Analysis and discussion should be provided of whether sufficient data remained to make calculations and inferences at night. The figure should be moved to an appendix/supplement, however.

**Response 8:** Due to night time data losses (mainly because of very small u* and large footprint) this paper focuses more on daytime events.

**Comment 9:** page 4 line 8 – I do not understand "Data from north and south systems were combined and selected based on the climatological footprint". Please explain further.

**Response 9**: In this study, we installed two EC systems at one flux tower minimize the gaps in the data due to maintenance and calibration instruments malfunction and accommodates seasonal changes in changing in wind direction. One tower facing north direction and another one facing south direction. The systems can cover all different angles of the study area which means more high-quality data were captured based on the prevailing wind that was coming from all different directions throughout the year. The south system facing south covers the angle from 90° to 270° and the rest of the area is covered by the north system facing north. The data from these two systems were combined and filtered based on the footprint analysis where only areas with Spartina alterniflora is studied.

**Comment 10:** page 4 line 9 – I do not understand "Only measurements that contributed to more than 70% of the $CO_2$ flux within the study area were used". Please explain further.

**Response 10:**

We used Kormann and Meixner (2001) in our footprint calculation. This method provides the cumulative source contribution (CSC) of the study field and the surrounding areas expressed in terms of percentage. 70% CSC implies that the distance from the measurement point contributes 70% of the observed flux. In our study, only data that fulfill the 70% CSC was retained.

**Comment 11:** Figure 6 not needed.

**Response 11:** Noted

**Comment 12:** Figure 7 not needed.

**Response 12:** Noted

**Comment 13:** Figure 9 is not needed; also see Major Comments C and D, above, regarding related confusion as to what the study actually did.

**Response 13:** Noted

**Comment 14:** Figure 10 is impressive and demonstrates the incredible volume of interesting data collected by the study team. However, see Minor Comment number 8 – I wonder a bit at the small standard deviations reported for nighttime NEE values given that the sample size after QA/QC was quite small for night times. The plot is very similar to that by Kathilankal et al., 2008 that spanned May through October, although this manuscript helpfully expands the figure through all 12 months.

**Response 14:** We believe this figure helps to understand the $CO_2$ exchange pattern at the study site. The study site growing season is almost a full year. We will look into the standard deviations as mentioned above although we are confident with how we filtered and processed our data to come out with good quality ones.

**Comment 15:** Figure 11 appears nearly identical to the kind of data presented in Kathilankal et al., 2008 and in Moffett et al. 2010. What is the new scientific insight added by this study that warrants re-publishing a known phenomenon?

**Response 15:** We found that figure 11 which similar to both Kathilankat et al., 2008 and Moffett et al. 2010 could give a very good view on how $CO_2$ exchange change diurnally and throughout the year.

**Comment 16:** Sections 3.2.1 and 3.2.2 – The manuscript to this point has not made it clear to me why we should be interested to compare May and October data, and so I do not see the point of these sections or Figure 12 or 13. Recommend omitting.

**Response 16:** The comparison was made for neap and spring tide events. We will reconsider to omit the results (as suggested). Otherwise we will add more details to it.

**Comment 17:** Page 8 line 18-19. This manuscript writes "Site studies of these authors are dominated by marsh grass species which grow upright, either Spartina alterniflora (Kathilankal et al., 2008) or Spartina foliosa and Distichlis spicata (Forbrich and Giblin, 2015; Moffett et al., 2010)." This is a direct quote – actually a mis-quote – of Forbrich and Giblin 2015, who wrote (page 1835) "Sites studied by these authors are both dominated by marsh grass species which grow upright, either Spartina alterniflora [Kathilankal et al., 2008] or Spartina foliosa and Distichlis spicata [Moffett et al., 2010]." but also clarified that "At our site, Spartina patens often lies prostrate forming a dense, green carpet. . ." (hence the mis-quote). [And actually the site by Moffett

et al. was as much Salicornia virginica as Spartina and Distichlis; west-coast US marshes are odd compared to east.]

**Response 17:** We apologize that we overlooked this sentence which at first it was put there for author's reference and intended to be restructured. We will amend this sentence in the revised version of the manuscript.

**Comment 18:** If use of a digital online supplement is enabled by the journal, the figures to be removed could be provided in a supplement.

**Response 18:** Noted.

---

## Author Comment (AC3) · 15 Dec 2017

**Response to Referee Comment 1 (RC1)**

Thank you for the constructive feedback on our manuscript. The comments and suggestions really help us to improve our manuscript. Please find our response to each of the comment below. We quote each of the comment and numbered it accordingly (i.e Comment 1) and our response (i.e Response 1) is right after the comment. Thank you.

**Major comments:**

**Comment 1:** The authors argue that with future sea level rise (and subsequent prolonged inundation), $CO_2$ net uptake might be lower in the future and the marsh will convert into a mudflat. I disagree with this: Many studies have shown that – while biomass production is important – it is not the only driver for the long-term stability of salt marshes with regard to sea level rise. Mostly it will depend on interactions between factors such as biomass production, sediment availability, tide range, rate of sea level rise as well as the possibility to transgress further inland (e.g. Morris et al. 2002, Kirwan et al. 2010, Kirwan et al. 2016).

**Response 1:** Thank you for pointing this out. In this study, we only report the short-term period of the data and discuss the data based on our findings. We do agree that biomass is not the key factor for the long-term stability of salt marshes and there are complex interactions between several factors. However, study on salt marsh is time-scale dependent. Depending on how long the model or prediction take place, the game could change. We will look into this matter and take into account the long-term effect on the ecosystem based on the existing reference.

**Comment 2:** My impression is that the description of the approach and the results are contradictory: From the results (Section 3.3, 3.4, Fig. 14, Tab. 4) I take it, that the August $CO_2$ fluxes were grouped in three classes based on the tide ratio and a light response curve was fitted to them separately as well as to 'non-flooded' conditions. This is not how I understood the methods (Section 2.4): I expected the light response curve to be fitted only to the $CO_2$ fluxes under non-flooded conditions (to get a reference value for non-flooded conditions). Afterwards, the modelled fluxes would be subtracted from the observed fluxes - independently from the tide ratio – to quantify the flux reduction. In a revised version of the manuscript, this should be explained better. It is not clear to me why the light response curve is fitted to $CO_2$ fluxes measured during partially or completely submerged conditions. I thought, the data coverage is so good that you know the magnitude of the 'real' fluxes, but you need to estimate how large they were if there was no tidal flooding (thus the reference value). Subsequently, I am not sure how to interpret the values for Fmea and Fmod in Tab. 4. Especially since the time series is not continuous (since the night time data are not used), I think the only time period that give us reasonable information is each single daytime flooding event. Thus, I suggest that the difference between Fmea and Fmod (only determined for non-flooded conditions) be calculated for each single daytime tide event and grouped according to tide ratio afterwards.

**Response 2:** Yes, we only use equation under non-flooded conditions to get the fluxes reference value for non-flooded conditions. Then we used this model to find fluxes reduction by subtracting modelled fluxes from the observed or measured fluxes which was done for each single daytime flooding event the. The values in table 4 was derived from this subtraction that was grouped into

different tide ratio. The reason why we separate the tide ratio into separate different class (partially or completely) was because, we could see how different level of flooding would impact the fluxes. We will explain in detail on this method in the revised version of the manuscript.

**Comment 3:** The comparison of neap and spring tide conditions in May and October is only descriptive and not connected to the fitting approach. I suggest to using the fitting approach for each month of the year and use these selected days to demonstrate the approach described above.

**Response 3:** In the revised version, we will add more months to give better explanation on our approach and use selected days as suggested.

**Minor comments:**

**Comment 1:** Page 1 ll. 22-25: Are all these numbers from the Chmura paper? Otherwise, they need references.

**Response 1:** We will insert the citation in the revised version.

**Comment 2:** Page 2 ll. 2 delete 'of'

**Response 2:** Deleted

**Comment 3:** Page 2 ll. 10-10-13: see comments above: There are biogeomorphic feedbacks between vegetation cover, tidal inundation and accretion rates, that are not directly linked to instantaneous $CO_2$ exchange but help marshes to keep their position relative to mean sea level.

**Response 3:** Noted. We will look further into it.

**Comment 4:** Page 2 ll. 29: I would rephrase that, do you 'hypothesize' this rather than 'believe'?

**Response 4:** Amended

**Comment 5:** Page 2 ll30: delete 'also'

**Response 5:** Deleted

**Comment 6:** Page 3 ll 6-9: Can you mention the height differences of the tall, medium and short plants? Which one do you use for the tide ratio? Also, how much variation is there during the entire growing season?

**Response 6:** We used the average height (tall, medium and short) for our tide ratio calculation. We will address it in detail in the revised version. We will also show the variation of plant height for the entire growing season by using monthly average plant height values.

**Comment 7:** Page 3 ll12-14: You do not need to say here that tides affect $CO_2$ exchange greatly, just mention the tide range.

**Response 7:** Noted.

**Comment 8:** Page 3 ll23: Are the tide heights reported in NAVD88 or relative to surface?

**Response 8:** The tide heights is in NAVD88

**Comment 9:** Page 4 ll. 7: Which quality control steps were applied?

**Response 9:** The quality control involved in fluxes calculation are stated in Page 3 and 4 sub-section 2.3. We will explain in detail the quality controls that we used in our flux calculation

**Comment 10:** Page 4 ll18-25: See comments above

**Response 10:** Noted. Explained above.

**Comment 11:** Page 4 ll26 – page 5 ll 5: Considering the high quality of the data set, I am surprised that you pick only one month and a couple of days to assess the tidal influence. The data coverage especially during the day is high and it would be possible to do this over the entire year and not only restrict yourself to the same climatic conditions (i.e. high irradiation).

**Response 11:** As mentioned above, we will provide more data (monthly and selected days) in the revised version. We purposely find the same climatic condition which are clear sky and similar irradiation intensity, to eliminate the effect cloud. We will consider providing mo

**Comment 12:** Page 5 ll 14 and ll 20-21: Contrary to these statements, Fig 8 shows that the marsh surface IS flooded during spring tide?!

**Response 12:** Thank you for pointing this out. We admit that we overlooked these sentences and will amend them accordingly. There was no flood during neap tide days and flooded during spring tide days for both months as clearly shown in Figures 7 and 8.

**Comment 13:** Page 6 ll 2-8: Most of this is descriptive and shown in Fig. 10 anyway. However, the observation that plants suffered from heat stress in July and August is interesting and would merit more analysis and discussion.

**Response 13:** We will provide more insight on this condition in the revised version.

**Comment 14**: Page 6 ll 14 – Page 7 ll5: See comments above

**Response 14:** Noted.

**Comment 15:** Page 7 ll 7 – 11: See comments above

**Response 15:** Noted.

**Comment 16:** Page 7 ll12 – 18: Why do you compare two random days (September as opposed to May, October or August as previously used) to give an example for the flux reduction instead of describing the results from the fitting procedure?

**Response 16:** As mentioned in section 2.4 page 4 l. 26 – 26 and page 5 l. 1 – 5, only days with clear sky condition during spring tide and neap tide days were used. There were very limited days with such condition. Therefore, we only able to use days in May and October for neap and spring tide comparisons in our study. The two different months represent the comparison between neap and spring tide days. Meanwhile, August data was used to quantitatively estimate monthly $CO_2$ reduction. These are the two main objectives in our study.

**Comment 17:** Page 7 ll 21-24: I think this should go into 'results'.

**Response 17:** We feel that it is fit to mention it in the discussion (sub-section 4.1) because it give some idea what we want to discuss after that.

**Comment 18:** Page 9 ll 2-5: See comments above: $CO_2$ exchange might be reduced instantaneously during inundation but that cannot be extrapolated over long periods of time.

**Response 18:** We admit that our study only focusses on short term periods of time. However, the monthly fluxes reduction (daytime) was obtained from modelled and observed instantaneously fluxes, then was summed up to get the total daytime reduction for that month.

**Figures and Tables:**

**Comment 1**: Fig. 1 and 4 are not really necessary.

**Response 1:** Noted.

**Comment 2:** Fig. 3 and Fig. 6 could be combined.

**Response 2:** Will be combined in the revised manuscript.

**Comment 3:** Fig. 11: This would work better with days that have been analyzed or discussed before (e.g. May/October).

**Response 3:** We will reconsider this in our revised manuscript.

**Comment 4:** Fig. 12 and 13 are not really necessary.

**Response 4:** We will consider removing them in the revised manuscript.

**Comment 5:** Fig. 14 needs more explanation: E.g., the different symbols are not explained, only the fit.

**Response 5:** Noted. We will explain more in the revised version of the manuscript

**Comment 6:** Table 1: All the values are given in the text, so this table is a repetition. Either change the text or remove the table.

**Response 6:** Noted. We will amend it in the revised version of the manuscript

**Comment 7:** Table 4: See comments above.

**Response 7:** Noted. Explained above